# Adjoint propagation of error signal through modular recurrent neural networks for biologically plausible learning

Zhuo Liu[1], Hao Shu[1], Linmiao Wang[1], Xu Meng[1], Yousheng Wang[1], Xuancheng Li[1], Wei Wang[2], Tao Chen[1]*

[1]School of Microelectronics, University of Science and Technology of China, Hefei, China; [2]Digital Intelligence Centre, Shenzhen Power Supply Bureau, China Southern Power Grid, Shenzhen, China

**Abstract** Biologically plausible learning mechanisms have implications for understanding brain functions and engineering intelligent systems. Inspired by the multi-scale recurrent connectivity in the brain, we introduce an adjoint propagation (AP) framework, in which the error signals arise naturally from recurrent dynamics and propagate concurrently with forward inference signals. AP inherits the modularity of multi-region recurrent neural network (MR-RNN) models and leverages the convergence properties of RNN modules to facilitate fast and scalable training. This framework eliminates the biologically implausible feedback required by the backpropagation (BP) algorithm and allows concomitant error propagation for multiple tasks through the same RNN. We demonstrate that AP succeeds in training on standard benchmark tasks, achieving accuracies comparable to BP-trained networks while adhering to neurobiological constraints. The training process exhibits robustness, maintaining performance over extended training epochs. Importantly, AP supports flexible resource allocation for multiple cognitive tasks, consistent with observations in neuroscience. This framework bridges artificial and biological learning principles, paving the way for energy-efficient intelligent systems inspired by the brain and offering a mechanistic theory that can guide experimental investigations in neuroscience.

*For correspondence:
tchen@ustc.edu.cn

Competing interest: The authors declare that no competing interests exist.

## Editor's evaluation

This article addresses a particular problem of current theories of supervised learning in cortical hierarchies: the fact that the inference signal afferents are not targeted reciprocally by potentially error-related connections. The authors propose an important idea where such a mismatch can be bridged by recurrent connections. The theory is supported by solid theoretical simulations.

## Introduction

A commonality between biological and artificial intelligence is their capability to adapt internal states based on external stimuli and responses. In modern artificial neural networks (ANN), this adaptation is predominantly governed by the backpropagation (BP) algorithm (*Rumelhart et al., 1986*; *LeCun et al., 2015*). BP has enabled performances surpassing humans in cognitive tasks such as natural language processing (*Vaswani et al., 2017*; *Huang et al., 2024*), pattern recognition (*He et al., 2016*), and mathematical reasoning (*Davies et al., 2021*; *Liu et al., 2025*). It minimizes the error between the actual and desired outputs of training datasets using the gradient descent method. Due

to the analytical nature of ANN models, the gradient of a loss function with respect to its learnable parameters can be computed via the chain rule of differentiation. This rule effectively propagates error signals backwards from the output layer back to preceding layers, hence the term *backpropagation*. However, an algorithmically plausible computation may be biologically and physically implausible. There is no conclusive evidence that biological neural networks backpropagate error signal through the same forward pathways, prompting further explorations of learning mechanisms more consistent with neurobiological observations (*Bengio, 2014*; *Bengio et al., 2016*; *Nøkland, 2016*; *Guerguiev et al., 2017*; *Whittington and Bogacz, 2017*; *Murray, 2019*; *Song et al., 2020*; *Ernoult et al., 2022*; *Greedy et al., 2022*; *Ororbia et al., 2023*; *Cheon et al., 2024*; *Max et al., 2024*). Algorithms like feedback alignment (FA) solve the weight transport problem (*Lillicrap et al., 2016*). However, a general principle of circuit topology that ensures credit assignment is lacking. Certain cortical neurons involved in stimulus representation receive no direct reciprocal signals from higher order circuits for learning (*Larkum et al., 1999*; *Larkum, 2013*; *Doron et al., 2020*; *Payeur et al., 2021*; *Young et al.,*

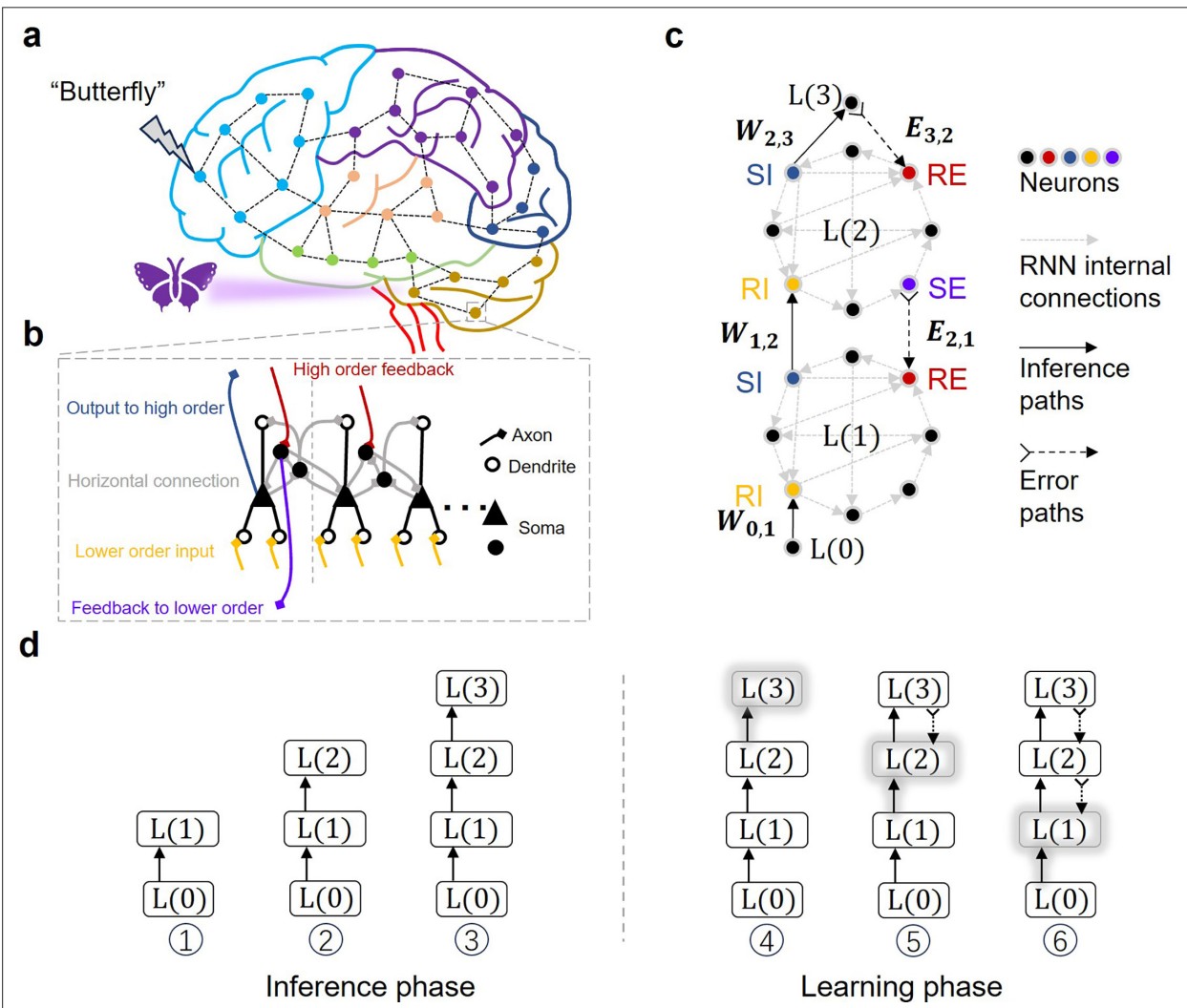

**Figure 1.** Schematics of the MR-RNN and the AP framework. (**a**) Illustration of cross-region recurrent connections in the brain. Each filled circle represents a local cortex area. (**b**) Illustration of horizontal and layer-wise recurrent connections in a cortical area. (**c**) An MR-RNN model with two modular RNNs $L(1)$ and $L(2)$. In this model, the internal connections of RNNs are sparse. For visual clarity, each circle represents a group of neurons, and the arrows represent bundles of interconnections. (**d**) The inference phase (left half) and the learning phase (right half) of AP. ①, The input data at $L(0)$ clamps $L(1)$ through $W_{0,1}$. ②, $L(1)$ in turn clamps $L(2)$ through $W_{1,2}$. ③, $L(2)$ produces an output at $L(3)$ via $W_{2,3}$. ④, The prediction error generated in the output layer $L(3)$ learns $W_{2,3}$. ⑤, The error of $L(3)$ layer nudges $L(2)$, producing local error at $L(2)$ layer for learning $W_{1,2}$. ⑥, Subsequently, $L(2)$'s error nudges $L(1)$ and yields local error for learning $W_{0,1}$. The shadows on the layers and arrows denote error computation and weight updating, respectively.

2021; *Moberg et al., 2025*). The link between local circuit motifs and network-wide credit assignment remains elusive.

In this work, we present an adjoint propagation (AP) framework for biologically plausible learning, grounded in the multi-region recurrent neural network (MR-RNN) model of the brain. Neurobiological evidence shows that most cortical regions exhibit rich feedback connections. Different functional areas are also extensively interconnected (see *Figure 1a–b*; *Markov et al., 2013*; *Park and Friston, 2013*; *Seguin et al., 2023*; *Dorkenwald et al., 2024*; *Rajimehr et al., 2024*), although the computational role of these connections is yet to be fully understood (*Markov et al., 2013*; *Yamins and DiCarlo, 2016*; *Krotov, 2021*; *Young et al., 2021*). Given the multi-scale recurrent connectivity, biological neural networks can be described by the MR-RNN model (see *Figure 1c*; *Richards et al., 2019*; *Perich and Rajan, 2020*; *Yang and Molano-Mazón, 2021*; *van Holk and Mejias, 2024*), which provides a useful tool for studying the complex brain dynamics (*Mejias et al., 2016*; *Joglekar et al., 2018*). Our AP algorithm leverages the recurrent connection in MR-RNN to propagate error signals without proprietary feedback paths for each neuron. We demonstrate that the error signal can be transmitted alongside the inference signal through the same RNN modules, enabling successful learning on benchmark classification tasks including MNIST (Modified National Institute of Standards and Technology) dataset of hand-written digits, FMNIST (Fashion MNIST) dataset of fashion items, and CIFAR-10 (Canadian Institute for Advanced Research 10) dataset of 10 classes of natural images. The adjoint propagation of error and inference signals allows dynamic allocation of neural resources for multiple cognitive tasks and supports reusing neurons for different functions (*Rigotti et al., 2013*; *Tye et al., 2024*). By incorporating the brains' structural properties, the AP framework not only bears explanatory power for biological intelligence, but also facilitates efficient learning in biologically inspired computing systems and adaptive robotics.

## Results

### Learning mechanism of adjoint propagation

To elaborate on the AP framework, we begin with a simple MR-RNN model shown in *Figure 1c*. The model consists of two modular RNNs $L(1)$ and $L(2)$, whose neurons are sparsely connected with directional paths internally. The sparsity and asymmetry of their internal connections reflect the characteristics of biological neural networks (*Bullmore and Sporns, 2012*; *Markov et al., 2013*; *Shapson-Coe et al., 2024*). The weight matrices for these internal connections, $W_1$ and $W_2$, are randomly initialized and kept constant. We use feedforward paths and feedback paths to connect the two RNN modules, represented by weight matrices $W_{1,2}$ and $E_{2,1}$, respectively. Just like in the brain, there are always two-way connections between different regions (*Sherman and Guillery, 2011*; *Markov et al., 2013*). For ease of discussion, we name four blocks of neurons in each RNN according to their connectivity, receiving inference signal (RI), sending inference signal (SI), receiving error (RE), and sending error (SE). The abbreviations, SI, RI, SE, and RE, only discriminate the roles of neurons in a specific computational task. The neurons in an RNN do not necessarily take one of the roles and do not need to be fundamentally different. The size of each block only depends on the connectivity of the network. By default, each functional block accounts for a quarter of the total neurons in its RNN module for initial demonstrations, until otherwise specified in later experiments. We assume that the SE neurons can compute the temporal difference of neural activities. Whether or not such computing is dispensable for the brain is presently unknown (*Payeur et al., 2021*; *Greedy et al., 2022*), but this assumption is supported by biological evidence of specialized neurons, the apical dendrites of pyramidal neurons and local circuit motifs (*Silberberg and Markram, 2007*; *Kubota, 2014*; *Leinweber et al., 2017*; *Roelfsema and Holtmaat, 2018*; *Sacramento et al., 2018*). A similar assumption is adopted by other learning frameworks, such as target propagation (*Bartunov et al., 2018*; *Lillicrap et al., 2020*; *Ernoult et al., 2022*) and local representation alignment (*Ororbia and Mali, 2019*; *Ororbia et al., 2023*).

The network operates in two phases, namely an inference phase and a learning phase. These two phases correspond to the two naturally dissected stages of learning, the presentation of stimulus and the synaptic adaptation according to credit assignment. In the inference phase (left half of *Figure 1d*), $L(1)$ receives data from the input nodes $u_0$ through the feedforward synaptic connection $W_{0,1}$. Under the influence of input signal, $L(1)$ evolves its states for $t_e$ steps to reach $u_1^{0,t_e}$, where $t_e$ represents a sufficiently large number. The superscript '0' in $u_1^{0,t_e}$ denotes the inference phase. Then $L(1)$ passes

information to L(2) through $W_{1,2}$. Upon receiving the feedforward signal, L(2) also updates for $t_e$ steps and transfers information to the output nodes through synaptic connections $W_{2,3}$, producing the output with an activation $f_3\left(u_3^0\right)$, where $u_3^0 = W_{2,3} \cdot f_2\left(u_2^{0,t_e}\right)$. In the learning phase (right half of *Figure 1d*), the error between the actual output and the target $y$, $e_3 = f_3\left(u_3^0\right) - y$, is fed to L(2) through weight matrix $E_{3,2}$, which nudges L(2) to slightly different states $u_2^{1,t_e}$, again for $t_e$ steps. The superscript '1' in $u_2^{1,t_e}$ denotes the learning phase. Note that the forward inference signal from L(1) is kept in the learning phase. The displacement of neural states $e_2 = f_2\left(u_2^{0,t_e}\right) - f_2\left(u_2^{1,t_e}\right)$ represents L(2)'s local error information, which trains $W_{1,2}$.

L(2) can relay the temporal error signal $e_2$ to L(1) through weight matrix $E_{2,1}$, thereby nudging L(1) into different states and producing an error signal to train $W_{0,1}$. The neural dynamics of L(1) and L(2) can be formulated as:

$$u_i^{\beta,t+1} = W_i \cdot f_i\left(u_i^{\beta,t}\right) + b_i + W_{i-1,i} \cdot f_{i-1}\left(u_{i-1}^{0,t_e}\right) - \beta E_{i+1,i} \cdot \left(e_{i+1}\right) \tag{1}$$

where $\beta = 0$ during the inference phase and $\beta = 1$ during the learning phase. Here, $i$ indexes the layers ($i = 0, 1, 2, 3$, including the input/output layer, and the RNN modules), $b_i$ denotes the bias vector, and $t$ indexes iteration step. $f_i$ represents the activation function of L(i) layer. For the output layer L(3), the activation function is SoftMax for classification tasks. For all other layers, it is the tanh function. At the input layer $u_0^{0,t_e} = u_0$ by definition, which corresponds to the input data. In matrices $W_{i-1,i}$ and $E_{i+1,i}$, only terms corresponding to existent connections from SI to RI or from SE to RE are non-zero.

The three feedforward synaptic connections $W_{0,1}$, $W_{1,2}$, and $W_{2,3}$ are learnable and can be updated according to the following rule:

$$\Delta W_{i,i+1} = -e_{i+1} \cdot f_i\left(u_i^{0,t_e}\right)^{\mathrm{T}} = f_{i+1}\left(u_{i+1}^{1,t_e}\right) f_i\left(u_i^{0,t_e}\right)^{\mathrm{T}} - f_{i+1}\left(u_{i+1}^{0,t_e}\right) f_i\left(u_i^{0,t_e}\right)^{\mathrm{T}} \tag{2}$$

The expanded items on the right-hand side represent a Hebbian item $f_{i+1}\left(u_{i+1}^{1,t_e}\right) f_i\left(u_i^{0,t_e}\right)^{\mathrm{T}}$ and an anti-Hebbian item $f_{i+1}\left(u_{i+1}^{0,t_e}\right) f_i\left(u_i^{0,t_e}\right)^{\mathrm{T}}$ as assumed in contrastive Hebbian learning (CHL). This assumption can be related to Long-Term Potentiation (LTP) and Long-Term Depression (LTD) of synapses (*Hyman et al., 2003*; *Scellier et al., 2018*). The error feedback weights $E_{i+1,i}$ need not adjust. The feedforward weights can align with the feedback weights through learning, as was found in previous studies (more detailed discussion on the algorithm in Appendix 1; *Lillicrap et al., 2016*).

## Spectral radius of modular RNNs and performance

To demonstrate AP's ability of credit assignment (see Materials and methods), we have used the MNIST and FMNIST datasets for training and test. For both datasets, we employed an MR-RNN model with two RNN modules of 1024 neurons, where each functional block contains 256 neurons. The weight matrices $W_1$ and $W_2$ are both randomly initialized with spectral radius (SR) of 0.25 (*Jaeger and Haas, 2004*; *Nakajima et al., 2024*), and the number of evolution steps is set to $t_e = 8$ (further discussion below). *Figure 2a–b* shows that the classification accuracies reach 97.47% and 88.97% for the two benchmark tasks (see more results in *Table 1*). Additionally, we trained the network to perform an archetypal neural task Two-Alternative Forced Choice (2AFC; see Appendix 2). As expected, the network shows robust performance in choosing the tilt direction of noisy images. While slightly under-performing comparably sized feedforward neural networks (FNN) trained by BP, the AP algorithm demonstrates effective credit assignment and successful training of the MR-RNN model.

To investigate how the spectral radius of RNNs affects classification performance and learning speed, we initialized $W_1$ and $W_2$ with SR values ranging from 0.01 to 2. We adopted the maximum Lyapunov exponent (MLE) $\lambda_{max}$ to quantify the dynamical properties of the RNNs (see *Figure 2c*; *Wolf et al., 1985*). MLE measures the divergence rate of two trajectories originating from infinitesimally close initial conditions. When the SR is less than 1, the MLE is negative, indicating that the RNNs resist small perturbations and converge to stable points (see the left two columns of *Figure 2d*). When SR is around 1, the MLE approaches zero, and the RNN exhibits marginal stability with quasi-periodic behavior (see the third columns of *Figure 2d*). When SR exceeds 1, the MLE becomes positive, and the RNNs become unstable, showing chaotic behavior (the right two columns of *Figure 2d*).

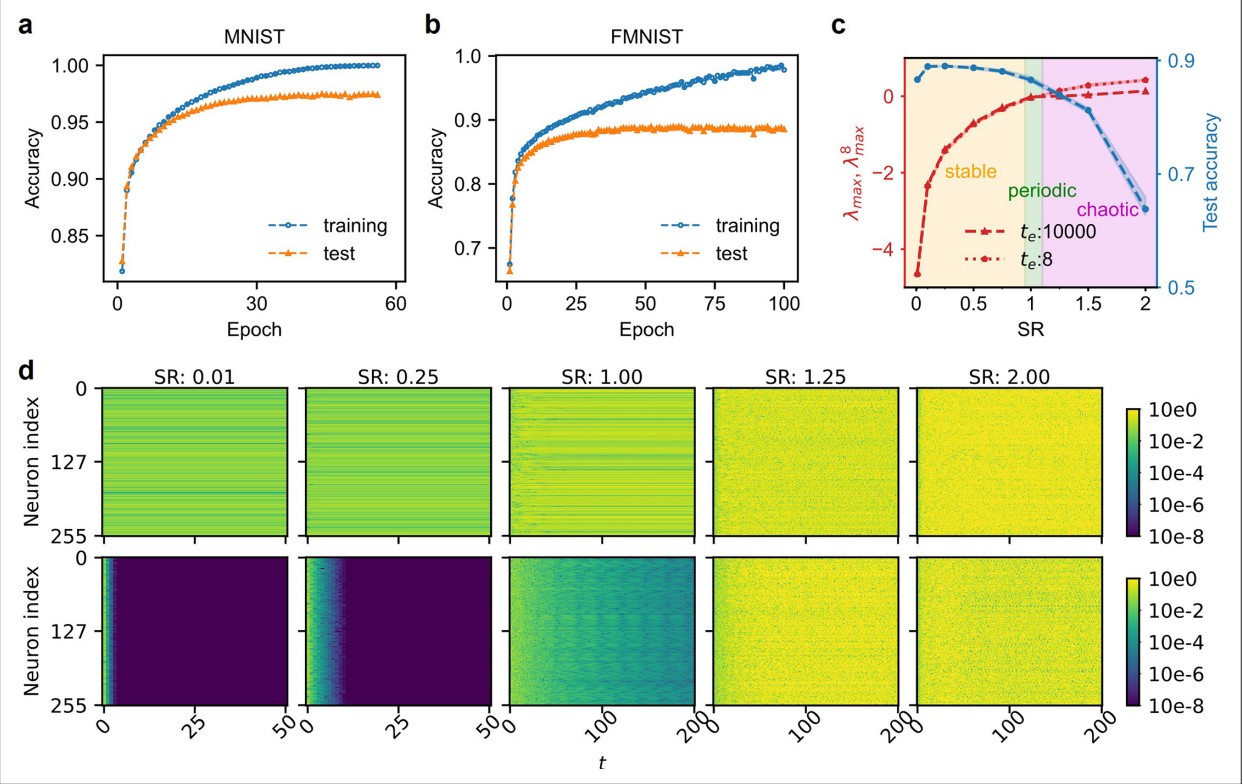

**Figure 2.** Learning performance of the AP framework. (**a–b**) The training and test accuracy on MNIST and FMNIST datasets. The increasing accuracies confirm the effectiveness of the AP algorithm in training the MR-RNN model. (**c**) MLE/FTMLE and accuracy on FMNIST versus SR. The accuracy peaks around SR of 0.1. SR affects the dynamics of the RNN and the accuracy on the datasets. MLE $\lambda_{max}$ and FTMLE $\lambda_{max}^8$ almost overlap with each other. Both increase when SR increases. The shaded area indicates three regions, namely stable, periodic, and chaotic dynamics of the RNN. (**d**) States evolution with different SR. The states of the network $u$ (top row) and the one-step difference $du$ (bottom row) are plotted against iteration step $t$. The one-step difference is the state of a step minus the state of its previous step, which reveals how much the states change.

The online version of this article includes the following source data for figure 2:

**Source data 1.** The source data for *Figure 2a–b*.

**Source data 2.** The source data for *Figure 2c*.

**Source data 3.** The source data for *Figure 2d*.

With small SRs, the RNNs converge in fewer than 10 iteration steps (left two columns of *Figure 2d*). To quantify the short-term dynamical properties of RNN, we chose the finite-time maximum Lyapunov exponent (FTMLE) instead (*Shadden et al., 2005*). Unlike MLE, which evaluates the convergence properties of trajectories over an infinite time span, FTMLE evaluates the divergence rate of two trajectories over a finite period (see Materials and methods; *Kanno and Uchida, 2014*; *Blessing Neamțu and Blömker, 2025*; *Storm et al., 2024*). As shown in *Figure 2c*, FTMLE with 8 iterations $\lambda_{max}^8$ closely approximates $\lambda_{max}$, which suggests 8 is a sufficiently large number of iterations for $t_e$. With further increase of iterations, the classification accuracy saturates and becomes solely determined by SR. Put differently, fewer iterations suffice.

*Figure 2c* shows that, when SR decreases to less than 0.1, the classification accuracy deteriorates. Although the systems remain stable with negative $\lambda_{max}$ below –2.5, the influence of input on the RNN dynamics rapidly fades. For SR values above 0.5, the RNNs take too long to converge, making them unsuitable for stationary input. The highest accuracy occurs in a broad SR range from 0.1 to 0.5, where $\lambda_{max}$ remains negative, ensuring the RNNs stay well within the stable regime and maintaining sufficient computational power. This observation is consistent with previous studies on RNN dynamics (*Nakajima et al., 2024*).

Similar to the equilibrium propagation (EP) algorithm, our AP framework represents error signals with the difference of neural states between an input-clamped inference phase and an error-clamped

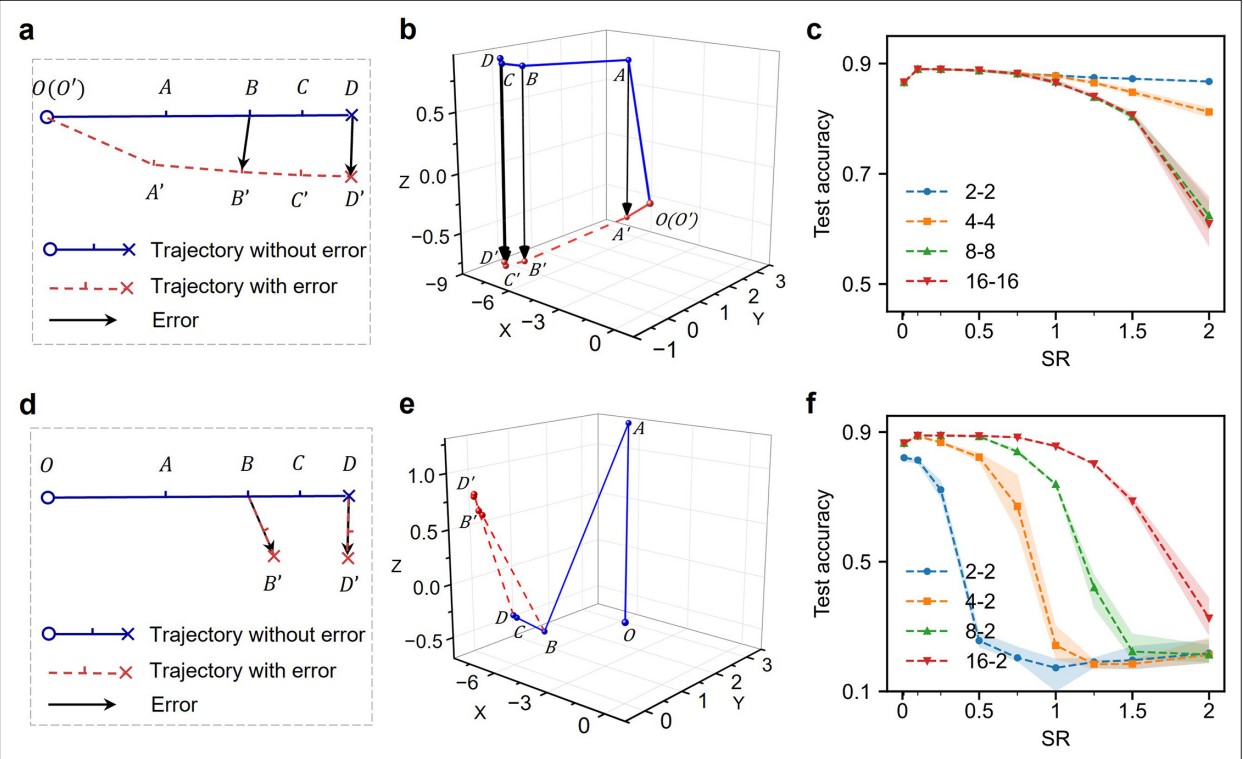

**Figure 3.** Error representations and fast learning. (**a**) Schematics of the short-trajectory error (STE). In the inference phase, an RNN evolves towards end point $D$ along the blue trajectory. We can terminate the neural evolution earlier and compute output error at intermediate steps ($A$, $B$, and $C$). Under the influence of its input and error, the RNN evolves from the same starting point $O$ and follows the red trajectory to $A'$, $B'$, $C'$, or $D'$. The vectors $AA'$, $BB'$, $CC'$, all approximate the end-point error $DD'$. (**b**) Visualization of STE trajectories. The STE trajectories of a 1024-node RNN for training on FMNIST dataset, visualized after PCA, correspond well to the schematics in a. (**c**) Test accuracy on FMNIST with different STEs. The legend $t_{e1}$-$t_{e2}$ specifies the numbers of iterations in the inference phase and the learning phase. Note that the first step is to feed the inference signal or error to an RNN, so $t = 2$ means the whole RNN states update once. The test accuracy for different STEs stays nearly the same for a small SR. For SR larger than 1, more iterations perform worse, because the neural state easily saturates and the inference and error signals are lost in the iterations. (**d**) Schematics of the error-perturbed error. The blue trajectory of the inference phase is the same as in a. However, if the output error is computed at an intermediate step (e.g. $B$) and lets the RNN continue updating with the error-perturbation, then the trajectory is steered towards destination $B'$. The vector $BB'$ also approximates the end point error.(**e**) Visualization of EPE trajectories. The actual EPE trajectories of a 1024-node RNN for training on FMNIST dataset, visualized after PCA, correspond well to the schematics in d. (**f**) Test accuracy on FMNIST with different EPEs.

The online version of this article includes the following source data for figure 3:

**Source data 1.** The source data for *Figure 3b, c, e and f*.

learning phase (*Scellier and Bengio, 2017*). However, unlike EP, which is grounded in multi-hidden layer recurrent neural networks and trains all the weights, AP trains only the feedforward connections between the RNN modules, excluding their internal weights. In our framework, error signals propagate adjointly with the inference signal via the Jacobians associated with the RE and SE nodes of RNNs, supporting modular and hierarchical learning (see below and Materials and methods). This approach takes full advantage of the fast and stable convergence of RNN modules (with suitable SR) to propagate both inference signal and error, thus favoring physical implementations, be it artificial or biological.

## Flexible error representation for fast and biologically plausible learning

As shown in *Equation 2*, the network needs to compute the error of post-synaptic neuron $e_i$ for learning. We found that the RNN modules do not need to reach equilibrium in the two phases to obtain the error. AP can approximate errors flexibly quasi-continuously (*Figure 3*). *Figure 3a* illustrates approximating the end-point error $DD'$ between two neural trajectories ($OD$ and $O'D'$; supposedly from the same initial conditions) by the error between two shorter trajectories (e.g. $AA'$ and $CC'$). We refer to these errors as short-trajectory errors (STE). We have projected a real high-dimensional

**Table 1.** Comparison of test accuracies among different training methods.
We use backpropagation and feedback alignment to train fully-connected feedforward neural networks with one or three hidden layers, each with 256 neurons. The training continues for a maximum of 300 epochs or terminates if the accuracy for the training set is reaching 99.99%.

| Model | Layer | MNIST | FMNIST | CIFAR10 |
|-------|-------|-------|--------|---------|
| | 1 | 98.04% ± 0.07% | 89.64% ± 0.09% | 53.66% ± 0.24% |
| BP | 3 | 98.14% ± 0.06% | 89.95% ± 0.21% | 53.65% ± 0.19% |
| | 1 | 97.94% ± 0.08% | 89.53% ± 0.11% | 52.67% ± 0.29% |
| FA | 3 | 97.77% ± 0.09% | 89.36% ± 0.04% | 52.29% ± 0.21% |
| | 1 | 97.47% ± 0.08% | 89.12% ± 0.06% | 50.11% ± 0.31% |
| AP | 3 | 97.30% ± 0.10% | 88.82% ± 0.19% | 50.06% ± 0.54% |

trajectory of RNN into three-dimensional space (*Figure 3b*) using principal component analysis (PCA) (*Hellton and Thoresen, 2017*; *Lever et al., 2017*). Indeed, the STE of the two phases stays nearly parallel at each iteration step ($AA'$, $BB'$, $CC'$), suggesting good approximation of the end-point error ($DD'$). Results in *Figure 3c* confirm that learning based on STE performs as well as learning based on end-point errors, which reduces the required number of iterations for RNNs and speeds up the training. It is worth pointing out that computing STE requires storing the output states of the first phase and restoring the initial conditions of RNNs for the inference phase. Despite the algorithmic speedup, whether there exists such an error representation with biological underpinnings is uncertain.

Alternatively, the RNN can advance along the inference trajectory first, and then we introduce the error from the higher-hierarchical layer to steer the RNNs away from trajectory. This perturbation produces a local error for the RNN, which we term error-perturbed error (EPE). The further the network progresses along the inference trajectory towards a stable point, the more the EPE becomes parallel to the end-point error (*Figure 3d and e*), leading to higher accuracy in the benchmark task (*Figure 3f*). Once the RNNs have converged under the clamping of input (stimulus kept online), computing EPE only requires one more step of iteration, which is local in time. Learning with EPE allows triggering the weight adjustments by error perturbation at the output layer. Therefore, AP accommodates quasi-continuous operation and needs no explicit coordination in physical substrates.

Thanks to its flexible error representations and the convergence properties of RNN modules, the AP algorithm enables fast and robust learning. We compared the learning speed of AP with equilibrium propagation on the FMNIST dataset in *Figure 4a*. EP and AP reach similar accuracy within the first few training epochs. However, as training progresses, the accuracy of EP declines as the weights continue adapting, likely due to the instability of its complex recurrent neural networks (*O'Connor et al., 2019*). In contrast, the performance of AP remains stable throughout the training, demonstrating remarkable robustness. Moreover, because the RNNs in the AP framework converge much faster, AP achieves comparable accuracy in less than half the training time of EP (*Figure 4b*), which requires significantly more iterations to converge (see Materials and methods). When scaled to deeper networks (*Figure 4c*), the advantage of AP in robustness and convergence speed becomes even more pronounced.

## Scalability of adjoint propagation

With the basic principle of AP established, we examine the scalability of AP in comparison with FA and BP (see *Figure 4d*). In shallow networks, the performance of AP is slightly lower than FA and BP. This performance gap likely results from the longer random feedback paths in AP, which cause larger deviation of gradient from BP. For the same number of layers, the errors in AP go through more synaptic connections than FA and BP, since an RNN module is treated as one layer in AP. As the number of layers increases, all three approaches lose performance due to known causes like gradient vanishing (explosion) or degradation problem (*He et al., 2016*). However, the problem of gradient vanishing (explosion) is severe for FA due to its fixed random feedback. In AP, the spectral radius of RNNs can

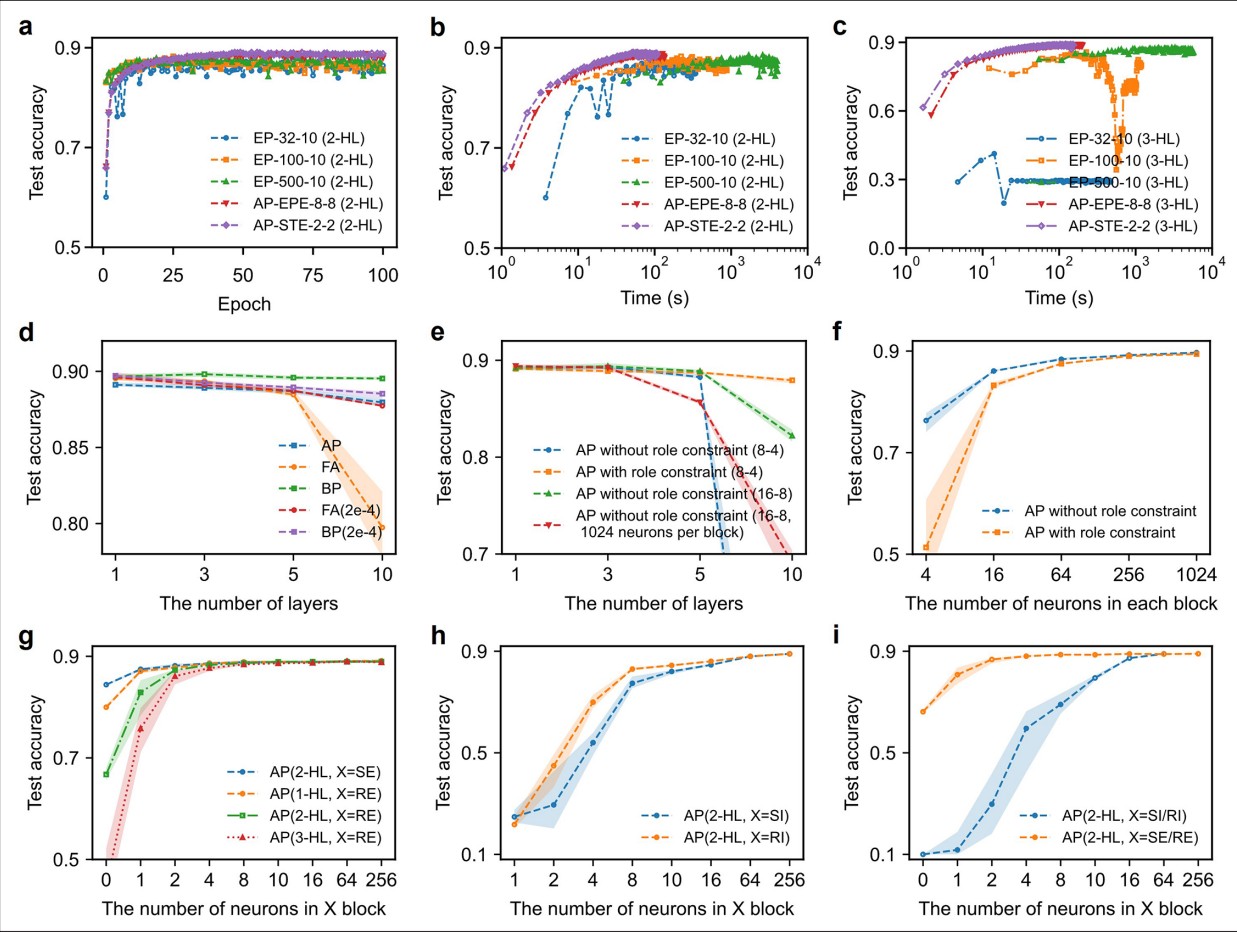

**Figure 4.** Comparison with EP and scalability. (**a**) Test accuracies of EP and AP with 2-hidden-layer (2-HL) on FMNIST versus training epochs. The numbers of iterations in the first/inference and second/learning phase are specified in the legend. EP learns faster for the first few epochs and exhibits signs of instability when iteration numbers are reduced to 32 and 10 for the two phases. In contrast, AP's accuracy is stable. (**b**) Test accuracies versus time for 2-hidden-layer (2-HL) networks. AP takes shorter time than EP to achieve the same accuracy. (**c**) Test accuracies versus time for 3-hidden-layer (3-HL) networks. The instability issue of EP becomes more severe when scaled to deeper networks, whereas AP's performance remains as stable as 2-hidden-layer networks. (**d**) Test accuracies on FMNIST of AP, FA, and BP with different number of layers. '2e-4' is the learning rate. (**e**) Test accuracies on FMNIST with or without role constraint versus the number of layers. '16–8' is the number of iterations in the first and second phase, which is sufficient for AP without role constraint. There are 256 neurons per block by default. RNNs without role constraints maintain the size of the shared block, thus the neural number of the layer is reduced by a factor of four. To compare the size effect, we also plot out the accuracy of unconstrained RNN with default neural number per layer 1024. (**f**) Test accuracies on FMNIST with or without sender/receiver constraint versus the number of neurons in each block. (**g–i**), Test accuracies on FMNIST versus the number of RE/RI/SE/SI neurons. 'X=SI/RI' means that the numbers of SI and RI neurons are both changed.

The online version of this article includes the following source data for figure 4:

**Source data 1.** The source data for *Figure 4*.

regulate feedback strength, hence offering an additional knob to tune the gradient amplitude and leading to better performance than FA (see also Appendix 3).

To study how the roles of neurons (SI, RI, SE, and RE) affect learning in AP, we compare the performance with and without role constraints (*Figure 4e*). Without the constraint, namely, all neurons receive and send both inference and error signals, the network needs more iterations to converge. The performance decreases faster in deeper structures without role constraint, because no role constraint means no regulation by spectral radius. We then study the performance scaling as a function of the lateral size of the network (*Figure 4f*). As expected, more neurons per layer results in better performance. However, the performance of the network with role constraint degrades more severely with utterly small RNNs, likely because modular RNNs with limited internal connections bottleneck the representation ability of a network.

In earlier experiments, we have assigned a quarter of the neurons in each RNN to send or receive error signals. To study how performance changes with varying number of RI/SI/RE/SE neurons, we fix the size of the network and change the size of each functional block. We have trained networks of 1–3 layers of RNNs, with their numbers of RE neurons ranging from 0 to 256. *Figure 4g* shows that the accuracy on the FMNIST dataset hardly decreases even when we decrease the numbers of RE neurons from 256 to just 10. Since the output is embedded in a 10-dimensional space, reducing the number of RE neurons to less than 10 complicates error representation and degrades performance. In case of no error feedback (0 RE neurons), the AP reduces to reservoir computing, which only trains the connections to the output (*Jaeger and Haas, 2004*; *Gauthier et al., 2021*; *Kleyko et al., 2025*). Results in *Figure 4g–i* suggest that decreasing neurons in the feedback path (SE/RE) has less effect compared to decreasing neurons in the feedforward path (SI/RI). After all, the inference-related (SI/RI) neurons need to represent high-dimensional input/features, whereas the error-related (SE/RE) neurons signal deviation from the targets that are inherently embedded in low-dimensional space.

In short, while AP initially underperforms FA and BP in shallow networks, its regulation by global property of RNN can mitigate the severe gradient issues of fully random feedback in FA. The distinct roles of neurons (SI, RI, SE, RE) turn out to facilitate stable and efficient learning in deep networks.

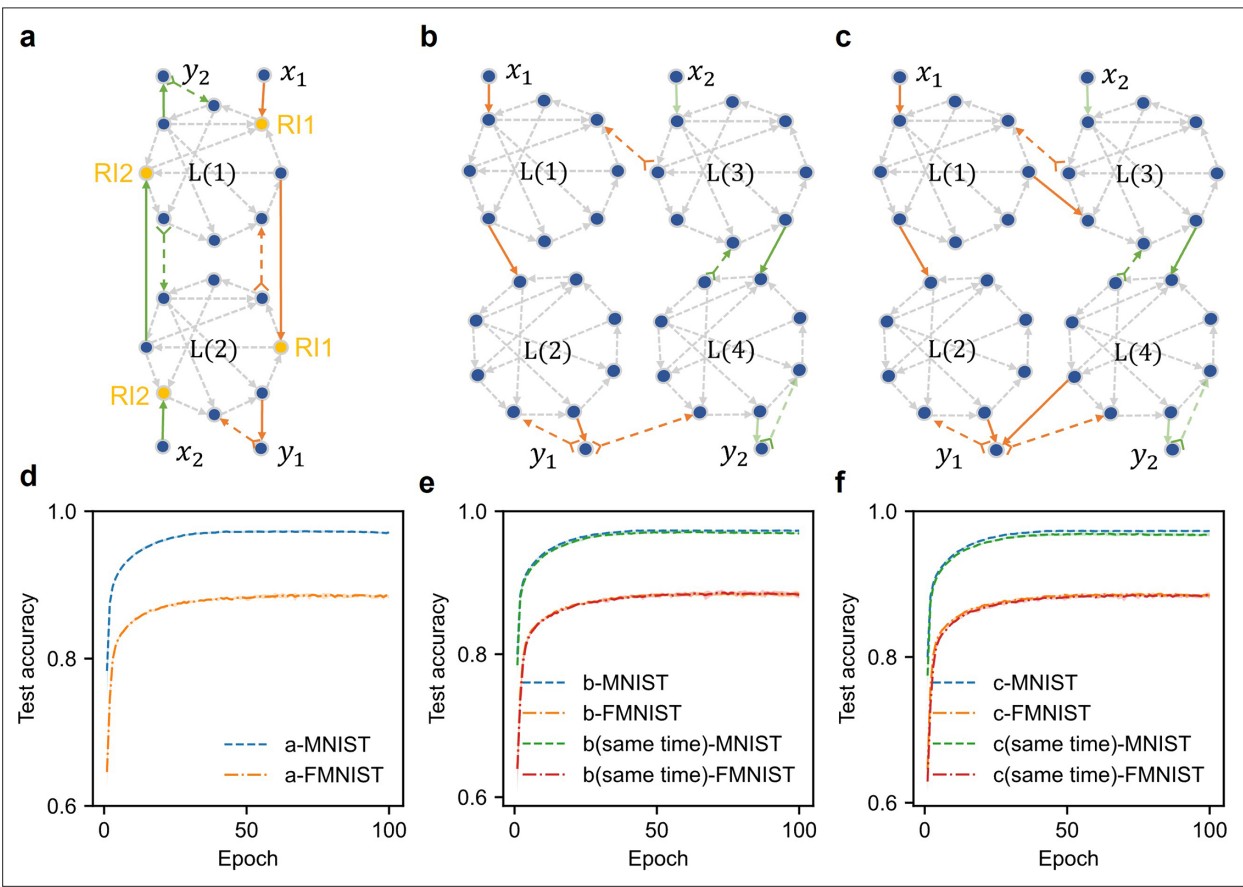

**Figure 5.** The reuse and flexible recruitment of neurons in AP. (**a**), Reuse of RNNs in different cognitive tasks. When the orange paths are used for the FMNIST task ($x_1$ for input and $y_1$ for output), the green paths are suppressed (grayed out in left panel), and vice versa for MNIST task (right panel, $x_2$ for input and $y_2$ output). The roles of neurons are different in different tasks. There are two different groups of RI neurons (RI1 and RI2) performing different tasks. (**b**) Flexible recruitment of neurons. Similar to **a**, the FMNIST task uses orange paths and the MNIST task uses green paths. Here, the FMNIST also shares the error feedback path between **L(3)** and **L(4)** (dashed green). (**c**) On the basis of **b**, FMNIST task can additionally recruit **L(3)** and **L(4)** for inference, namely incorporating the green path between **L(3)** and **L(4)** without learning it. The MNIST classification task only uses the green paths. (**d-f**) Accuracies of different tasks in different configurations illustrated in **a**, **b**, and **c**.

The online version of this article includes the following source data for figure 5:

**Source data 1.** The source data for *Figure 5d-f*.

The possibility of significantly reducing error-signaling (SE/RE) neurons enhances AP's computational appeal and biological plausibility, as it requires fewer dedicated feedback pathways.

## Dynamical resource allocation for multiple cognitive tasks

In biological neural networks, individual neurons can participate in multiple tasks, that is reuse of neurons for different functions, which is an important aspect of cognitive flexibility (*Rigotti et al., 2013*; *van Holk and Mejias, 2024*). The AP framework is inherently commensurate with multi-tasking of neurons. We can use the same RNN modules to perform both MNIST and FMNIST classifications, each task with its own subsets of forward and error feedback paths. As illustrated in *Figure 5a*, the FMNIST data flows through the orange forward paths and the two RNN modules to yield an output, meanwhile, the corresponding error signal propagates backwards through the dashed orange error paths and the same two RNN modules. Once the training is completed, we suppress the orange and dashed orange paths and use the green forward paths and dashed green error paths for MNIST training. The resulting accuracies are 97.35% for MNIST and 88.83% for FMNIST (*Figure 5d*), nearly identical to those obtained by training on separate networks. In principle, more tasks can reuse the same RNN modules, thus saving computational resources.

We note that sharing RNNs for different tasks requires selectively suppressing irrelevant pathways, which has concrete ground in neuroscience (*Gazzaley et al., 2005*; *Feldmann-Wüstefeld and Vogel, 2019*; *Kuan et al., 2024*; *Langdon and Engel, 2025*). Combined with selective suppression (termed configuration), the AP framework allows flexible recruitment of RNN modules and pathways for cognitive tasks. As shown in *Figure 5b*, four RNN modules are recruited to perform the MNIST and FMNIST classification tasks. Here, FMNIST uses feedforward path $x_1 - L(1) - L(2) - y_1$ (solid orange arrow) for inference, and $y_1 - L(4) - L(3) - L(1)/y_1 - L(2)$ for error propagation (dashed orange arrow). As for MNIST, the feedforward path is $L(3)$ (solid green arrow) and the feedback path is $y_2 - L(4) - L(3)$ (dashed green arrow). The two tasks share the error feedback path from $L(4)$ to $L(3)$. The network alternates between two configurations and learns for one epoch of the dataset under each configuration (see Algorithm 2 in Appendix 2). The accuracies reach 88.73% for FMNIST and 97.41% for MNIST dataset. We also carried out training on the two datasets at the same time, and the network is still able to learn even when the error signals of the two tasks propagate simultaneously along the $L(4) - L(3)$ path (see Algorithm 3 in Appendix 2). For one task, the signals of the other task can be seen as noise (more discussion in Appendix 4), which does not jeopardize learning (*Neelakantan et al., 2015*; *Duan et al., 2021*).

For the FMNIST task, we can additionally recruit $L(3)$ and $L(4)$ for inference (see *Figure 5c*), namely $x_1 - L(1), L(3) - L(2), L(4) - y_1$, without learning the green feedforward path between $L(3)$ and $L(4)$. The additional resource allocation results in 88.84% accuracy, meanwhile leaving the MNIST classification intact (*Figure 5f*). Again, the two tasks can be learned at the same time. $L(3) - L(4)$ can transfer the inference and error signals of the two datasets simultaneously, and the RNNs can multi-task. These experiments demonstrate AP's ability to flexibly recruit neurons for one task and to simultaneously propagate signals of multiple tasks, which are favored traits that contribute to resilience of both artificial and biological neural networks.

## Discussion

The MR-RNN model has been adopted in computational neuroscience to describe brain activities (*Perich and Rajan, 2020*; *van Holk and Mejias, 2024*), but its potential for artificial intelligence remains largely untapped. In this work, we bridge the gap by introducing a biologically plausible AP learning framework that unifies principles from equilibrium propagation, feedback alignment, and draws inspiration from the MR-RNN model. We demonstrate that RNNs with sparse and random internal connections—capable of inference tasks—can simultaneously propagate error signals via their Jacobians without structural modification. Moreover, by dynamically suppressing pathways, akin to cortical gating mechanisms, the AP enables sharing RNNs by multiple cognitive tasks and flexible recruitment of subnetworks for a single task. It sheds light on a long-standing question: how can sparse and modular networks support resilient cognitive functions?

Recent experimental evidence indicates distinct roles of cortical neurons (e.g. L5-IT and L5-ET neurons, which contribute distinctively to sensory processing and learning *Young et al., 2021*; *Moberg*

*et al., 2025*). Some neurons possess relatively sparse reciprocal cortical connectivity, which limits their direct access to feedback signals. As a result, they may rely on indirect routes—such as intracortical pathways or subcortical relays—to integrate information relevant for learning or credit assignment. In this context, the AP framework offers a possible mechanism by which effective learning signals can be conveyed even when direct reciprocal connections are limited.

The modular RNNs need not fine-tune their internal connections but only adjustments of global property like spectral radius. For static image classification tasks studied in this work, relatively smaller spectral radius (0.1–0.5) accelerates convergence and learning. Too large a spectral radius causes instability, and too small a spectral radius leads to attenuated network response. We anticipate that, for sequence tasks like voice recognition, other spectral radii may fare better. The global property adjustment of RNNs can be realized by neural modulators, such as serotonin and dopamine (*Peters et al., 2021*; *van Oostrum and Schuman, 2025*), which also allows selectively to recruit neural resources for different tasks. We stress that multi-tasking in AP differs from other conventional models (*Ruder, 2017*; *Thung and Wee, 2018*). A convolutional structure performing feature extraction for different image datasets can be treated as multi-tasking. However, such multi-tasking has to do with sharing feature representations (*Fang et al., 2017*). In contrast, the AP framework goes beyond to allow concomitant propagation of inference and error signals of different categories of tasks through the shared network, which is facilitated by the dynamical properties of RNNs.

The dynamical equations of RNNs are rate-coded approximations of spiking neural networks (SNNs, *Maass, 1997*). We expect the AP framework also works for SNN. SNNs feature event-driven dynamics, temporal sparsity, and non-differentiable spike events, all of which introduce challenges for gradient estimation and credit assignment (*Wu et al., 2019*; *Bellec et al., 2020*; *Yin et al., 2023*). AP offers an alternative option to solve the training issue of SNN. While the present study focused on static image classification, the RNNs can capture temporal features, which is critical for tasks such as speech recognition, video understanding, or closed-loop sensorimotor control. Extending AP to SNNs and temporal tasks is a lucrative direction for future study (*Murray, 2019*; *Bellec et al., 2020*; *Marschall et al., 2020*).

While its classification accuracies on benchmark datasets lag behind state-of-the-art deep learning, the framework is amenable to structural enhancements, such as adding residual connections and convolutional layers, for enhancing performance. (*He et al., 2016*; *Nøkland, 2016*; *Laborieux et al., 2021*; *Laborieux and Zenke, 2024*). Its compatibility with plastic feedback connections also opens up further potential for optimization (*Akrout et al., 2019*). More importantly, AP provides a blueprint for brain-inspired computing: its modular RNNs leverage sparsity and convergence properties that favor efficient hardware implementation. By reconciling neurobiological constraints with machine learning principles, AP advances both fields—from understanding cortical computation to designing adaptive, energy-efficient hardware for AI.

## Materials and methods
### Training details
We used three archetypal datasets of machine learning for demonstrating the effectiveness of AP, MNIST, FMNIST, and CIFAR-10 (*Krizhevsky, 2009*; *Xiao et al., 2017*). The datasets are provided by the open-source machine learning library Torch, and all pixels of samples in the datasets are scaled to [0,1]. We adopted batch learning in the experiments, a common approach in machine learning. We used a batch size of 500 in training by default and employed the adaptive moment estimation (Adam) optimizer for learning rate adaptation (see *Table 2*; *Kingma and Ba, 2015*). In physical hardware or

**Table 2.** The Adam optimizer parameters.

| Parameter name | Default value |
| --- | --- |
| Learning rate (MNIST FMNIST) | 0.001 |
| First-order moment estimation decay rate ($\beta_1$) | 0.9 |
| Second-order moment estimation decay rate ($\beta_2$) | 0.999 |
| Small constant for numerical stability ($\epsilon$) | 1e-8 |

biological networks, batch learning is non-local in time and demands storing the losses of all samples in a batch. However, even with a batch size of 1 (without Adam optimizer), the AP algorithm can reach 88.81% accuracy for the FMNIST dataset after 100 epochs, suggesting that a suitable physical substrate can be trained with the AP framework in a biologically plausible manner.

For network initialization, we followed *Nøkland, 2016* and sampled feedforward connections $W_{i,i+1}$ (between the SI neurons in L($i$) and RI neuron in L($i + 1$)) from a uniform distribution in the range $\left[-\frac{1}{\sqrt{N_{i,\mathrm{SI}}}}, \frac{1}{\sqrt{N_{i,\mathrm{SI}}}}\right]$, here, $N_{i,\mathrm{SI}}$ is the number of SI neurons in L(i) layer. For the error feedback matrices, the entries are sampled from [-1,1]. The rest of the entries are kept zero. Since RNNs with small spectral radius already provide scaling capabilities, the error weights are not scaled to avoid gradient vanishing. All biases are set to zero and fixed. The default connection rate of the RNNs, defined as the actual number of connections over the number of all-to-all connections, is 0.25.

## Maximum Lyapunov exponent and finite time maximum Lyapunov exponent

We adopted established methods for calculating the maximum Lyapunov exponent and finite time maximum Lyapunov exponent (*Wolf et al., 1985*; *Shadden et al., 2005*). Consider an $n$-dimensional discrete dynamical system with dynamics $u_{t+1} = F(u_t)$, whose spectrum of Lyapunov exponents is $\{\lambda_i, i = 1, \ldots, n\}$. A small disturbance $\delta u_t$ at state $u_t$ evolves in the following way:

$$\delta u_{t+1} = J(u_t)\ \delta u_t, \tag{3}$$

where $J(u_t) = \partial F(u_t)/\partial u_t$ is the Jacobian matrix at state $u_t$. We initialized the state vector $u_0$ randomly and a perturbation vector $p$ of unit length: $\|p\| = 1$. For discrete time steps $t = 1, 2, \ldots, N-1$, we run the following steps:

a. Update the system state: $u_{t+1} = F(u_t)$.
b. Calculate the Jacobian matrix: $J(u_t) = \partial F(u_t)/\partial u_t$.
c. Update the perturbation: $\delta u_{t+1} = J(u_t)p$.
d. Normalized perturbation vector: $p \leftarrow \frac{\delta u_{t+1}}{\|\delta u_{t+1}\|}$.

The MLE is $\lambda_{max} = \frac{1}{N}\sum_{t=0}^{N-1}\ln\left(\|\delta u_{t+1}\|\right)$ for large $N$. When $N$ is small, such as 8 in the main text, the result is the finite time maximum Lyapunov exponent.

## Principal component analysis for dimension reduction

Since the states of the RNNs comprise a 1024-dimensional vector, we have used principal component analysis method to reduce the dimension and visualize the trajectories (*Figure 3*). The trajectories of an RNN recorded in experiments are represented by a series of 1024-dimensional vectors, which can be arranged in a matrix of dimension $1024 \times t_e$, where $t_e$ is the number of iteration steps. PCA selects three eigenvectors of the covariance matrix of the trajectory matrix, which correspond to the three largest eigenvalues. These eigenvectors serve as principal axes, and the original data is projected onto them to obtain a $3 \times t_e$ matrix that represents the trajectory in a three-dimensional space.

## Comparison with equilibrium propagation

To compare with equilibrium propagation, we trained an RNN model with the same number of hidden layers as the MR-RNN model considered in the main text, 2 hidden layers. The EP algorithm follows the description in *Scellier and Bengio, 2017*. Note that each RNN module involved in AP is considered to be a hidden layer without learning of its internal weights. In contrast, the hidden layers involved in EP are neurons without lateral connections; it is the layer-wise two-way connections that make the whole network recurrent. For a fair comparison, the numbers of layer-wise feedforward connections and feedback connections in the two frameworks are kept the same. The number of neurons in an MR-RNN layer is four times that of the hidden layer in EP.

Trainings involved in comparison in *Figure 4a–b* have been performed on a NVIDIA RTX 4070 GPU. Results in other sections are obtained on NVIDIA RTX 4080 GPU or NVIDIA RTX A6000 GPU.

## Acknowledgements

We thank Danlei Bi and Aihui Tang for helpful discussions. We acknowledge financial support from the University of Science and Technology of China and the Chinese Academy of Sciences.

## Additional information

### Funding

| Funder | Grant reference number | Author |
|---|---|---|
| Chinese Academy of Sciences | | Tao Chen |
| University of Science and Technology of China | | Tao Chen |

The funders had no role in study design, data collection and interpretation, or the decision to submit the work for publication.

### Author contributions

Zhuo Liu, Data curation, Software, Formal analysis, Investigation, Visualization, Methodology, Writing – original draft, Writing – review and editing; Hao Shu, Linmiao Wang, Xuancheng Li, Wei Wang, Validation, Writing – review and editing; Xu Meng, Yousheng Wang, Validation, Investigation, Writing – review and editing; Tao Chen, Conceptualization, Resources, Formal analysis, Supervision, Funding acquisition, Validation, Investigation, Visualization, Methodology, Writing – original draft, Project administration, Writing – review and editing

### Author ORCIDs

Zhuo Liu ⓘ https://orcid.org/0000-0003-1508-6188
Hao Shu ⓘ https://orcid.org/0009-0004-1394-9108
Tao Chen ⓘ https://orcid.org/0000-0002-6768-3630

### Decision letter and Author response

Decision letter https://doi.org/10.7554/eLife.108237.sa1
Author response https://doi.org/10.7554/eLife.108237.sa2

## Additional files

### Supplementary files

MDAR checklist

### Data availability

All data generated or analysed during this study are included in the manuscript and supporting files. The code of the study is provided via GitHub at https://github.com/Zero0Hero/Adjoint-Propagation-framework (copy archived at *Liu, 2026*).

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

## Appendix 1

## Algorithm and formula derivation

### Algorithms of AP

The detailed learning procedure of AP is described with pseudocode in Algorithm 1. The experiments associated with *Figure 5* in the main text follow Algorithm 2-3. The RNN modules that are irrelevant in the inference phase are set to initial states (e.g. L(3) and L(4) for training on FMNIST in AP model of alternating configurations). In this case, the error-perturbation is directly added to the initial state in the learning phase.

---

Appendix 1—algorithm 1: AP model

---

**Input**: A batch of input and target $(x, y)$
**Parameter**: $\theta = \left(W_{0,1}, W_{1,2}, W_{2,3}, W_1, W_2, E_{2,1}, E_{3,2}\right)$
**Output**: $\theta$
1. **Function** Inference-phase $(\theta, x)$:
2. $u_0^{0,t_e} = u_0 = x$
3. **for** $i \leftarrow 1$ to 2 **do**
4. $u_i^{0,0} \leftarrow 0$
5. **for** $t \leftarrow 0$ to $t_e - 1$ **do**
6. $u_i^{0,t+1} \leftarrow W_i \cdot f_i(u_i^{0,t}) + W_{i-1,i} \cdot f_{i-1}(u_{i-1}^{0,t_e})$
7. **end**
8. **end**
9. $u_3^{0,t_e} = u_3 = W_{2,3} \cdot f_2(u_2^{0,t_e})$
10. $\Lambda_1 = [u_i^{0,t_e}], i = 1, 2, 3$
11. **return** $\Lambda_1$

12. **Function** Learning-phase $(\theta, \Lambda_1, y)$:
13. $e_3 = f_3(u_3) - y$
14. **for** $i \leftarrow 2$ to 1 **do**
15. $u_i^{1,0} \leftarrow 0$ if STE else $u_i^{0,t_e}$
16. **for** $t \leftarrow 0$ to $t_e - 1$ **do**
17. $u_i^{1,t+1} \leftarrow W_i \cdot f_i(u_i^{1,t}) + W_{i-1,i} \cdot f_{i-1}(u_{i-1}^{0,t_e}) - E_{i+1,i} \cdot e_{i+1}$
18. **end**
19. $e_i = f_i(u_i^{0,t_e}) - f_i(u_i^{1,t_e})$
20. **end**
21. $\Lambda_2 = [e_i], i = 1, 2, 3$
22. **return** $\Lambda_2$

23. **Function** Updating-Weights $(\theta, \Lambda_1, \Lambda_2)$:
24. $\Delta W_{2,3} \leftarrow -e_3 \cdot f_2(u_2^{0,t_e})^T$
25. $\Delta W_{1,2} \leftarrow -e_2 \cdot f_1(u_1^{0,t_e})^T$
26. $\Delta W_{0,1} \leftarrow -e_1 \cdot f_0(u_0)^T$

---

---

Appendix 1—algorithm 2: AP model of alternating configurations

---

*Continued on next page*

*Continued*

---

**Input**: Two datasets $(x_1, y_1), (x_2, y_2)$, their AP configuration and connections parameters $(G_1, \theta_1), (G_2, \theta_2)$, the number of training epochs and batch number for both tasks $(n_{epoch}, n_{batch})$
**Output**: $\theta_1, \theta_2$
1: **Function** Training $([(x_1, y_1), (G_1, \theta_1)], [(x_2, y_2), (G_2, \theta_2)])$:
2: **for** *epoch* $\leftarrow$ 1 **to** $n_{epoch}$ **do**
3:     Configuring network with $G_1$
4:     **for** *batch* $\leftarrow$ 1 **to** $n_{batch}$ **do**
5:         Inference-phase with $x_1$
6:         Training-phase with $(x_1, y_1)$
7:         Updating parameters $\theta_1$
8:     **end**
9:     Configuring network with $G_2$
10:     **for** *batch* $\leftarrow$ 1 **to** $n_{batch}$ **do**
11:         Inference-phase with $x_2$
12:         Training-phase with $(x_2, y_2)$
13:         Updating parameters $\theta_2$
14:     **end**
15: **end**
16: **return** $\theta_1, \theta_2$

---

**Appendix 1—algorithm 3**: AP model of simultaneous input

---

**Input**: Two datasets $(x_1, y_1), (x_2, y_2)$, their AP configuration and connections parameters $(G, \theta)$, the number of training epochs and batch number for both tasks $(n_{epoch}, n_{batch})$
**Output**: $\theta_1$
1: **Function** Training $([(x_1, y_1), (x_2, y_2), (G, \theta)])$:
2: **for** *epoch* $\leftarrow$ 1 **to** $n_{epoch}$ **do**
3:     Configuring network with $G$
4:     **for** *batch* $\leftarrow$ 1 **to** $n_{batch}$ **do**
5:         Inference-phase with $x_1, x_2$
6:         Training-phase with $(x_1, y_1), (x_2, y_2)$
7:         Updating parameters $\theta$
8:     **end**
9: **end**
10: **return** $\theta_1$

---

## Derivation of update rule

We can derive the weight update rule following the principle of discrepancy reduction (***Ororbia and Mali, 2019***), namely, reducing the discrepancy between neural states in the inference phase and learning phase. The argument is that when the network is well trained, the error should be small, and the discrepancy between the two phases is minimized. We define the total loss as:

$$\text{Loss}\,(\theta) = \sum_{i=1}^{L} k_i \mathcal{L}_{\text{i}}\left(u_i^{0,t_e}, u_i^{\beta,t_e}\right) = \sum_{l=1}^{L} k_i \left(\| f_i\left(u_i^{1,t_e}\right) - f_i\left(u_i^{0,t_e}\right) \|\right)^2. \tag{A1}$$

Where $\|\cdot\|$ denotes Euclidean norm. $k_i$ is a scalar that weights the contribution of a specific local loss to the total loss. Here $k_i = 1/2$. The error of each layer can be defined as the partial derivative of the loss to the activation:

$$
\begin{aligned}
e_i \;&=\; \frac{\partial \,\text{Loss}(\theta)}{\partial f_i\left(u_i^{0,t_e}\right)} \\[2mm]
&=\; \frac{\partial \sum_{l=1}^{L} k_i \left(\left\| f_i\left(u_i^{1,t_e}\right) - f_i\left(u_i^{0,t_e}\right)\right\|\right)^2}{\partial f_i\left(u_i^{0,t_e}\right)} \\[2mm]
&=\; \frac{\partial \left(\left\| f_i\left(u_i^{1,t_e}\right) - f_i\left(u_i^{0,t_e}\right)\right\|\right)^2}{2\partial f_i\left(u_i^{0,t_e}\right)} \\[2mm]
&=\; f_i\left(u_i^{0,t_e}\right) - f_i\left(u_i^{1,t_e}\right)
\end{aligned}
\tag{A2}
$$

Further, we can deduce the weight update rule from the gradients of the loss with respect to the weights:

$$
\begin{aligned}
\Delta W_{i-1,i} \quad &= -\frac{\partial \text{Loss}(\theta)}{\partial W_{i-1,i}} \\
&= -\frac{\partial \text{Loss}(\theta)}{\partial f_i\left(u_i^{0,t_e}\right)} \frac{\partial f_i\left(u_i^{0,t_e}\right)}{\partial u_i^{0,t_e}} \left(\frac{\partial W_{i-1,i} \cdot f_{i-1}\left(u_{i-1}^{0,t_e}\right)}{\partial W_{i-1,i}} + \frac{\partial \left(W_i \cdot f_i\left(u_i^{0,t_e-1}\right)\right)}{\partial W_{i-1,i}}\right)
\end{aligned}
\tag{A3}
$$

Since the spectral radius of $W_i$ is small, we assume that the second term in the bracket can be omitted, and rewrite the forward weight update rule in the following form:

$$
\begin{aligned}
\Delta W_{i-1,i} \quad &\approx -\frac{\partial \text{Loss}(\theta)}{\partial f_i\left(u_{(i)}^{0,t_e}\right)} \frac{\partial f_i\left(u_{(i)}^{0,t_e}\right)}{\partial u_i^{0,t_e}} \frac{\partial W_{i-1,i} \cdot f_{(i-1)}\left(u_{(i-1)}^{0,t_e}\right)}{\partial W_{i-1,i}} \\
&= -e_i \cdot f_{i-1}(u_{i-1}^{0,t_e})^T \odot f_i'(u_i^{0,t_e}) \\
&\approx -e_i \cdot f_{i-1}(u_{i-1}^{0,t_e})^T \\
&= -f_i\left(u_i^{0,t_e}\right) \cdot f_{i-1}\left(u_{i-1}^{0,t_e}\right)^T + f_i\left(u_i^{1,t_e}\right) \cdot f_{i-1}\left(u_{i-1}^{0,t_e}\right)^T
\end{aligned}
\tag{A4}
$$

where $\odot$ denotes element-wise multiplication. It is the same as the update rule of most discrepancy-based algorithms in conventional FNN. We can further drop the derivative of the activation function, because previous studies have shown that the derivative can be omitted as long as the activation function is monotonically non-decreasing (*Melchior and Wiskott, 2019*; *Ororbia and Mali, 2019*). The accuracies with and without $f'$ show negligible difference (*Appendix 1—figure 1*). Expanding the error term, we arrive at the last row of the equation.

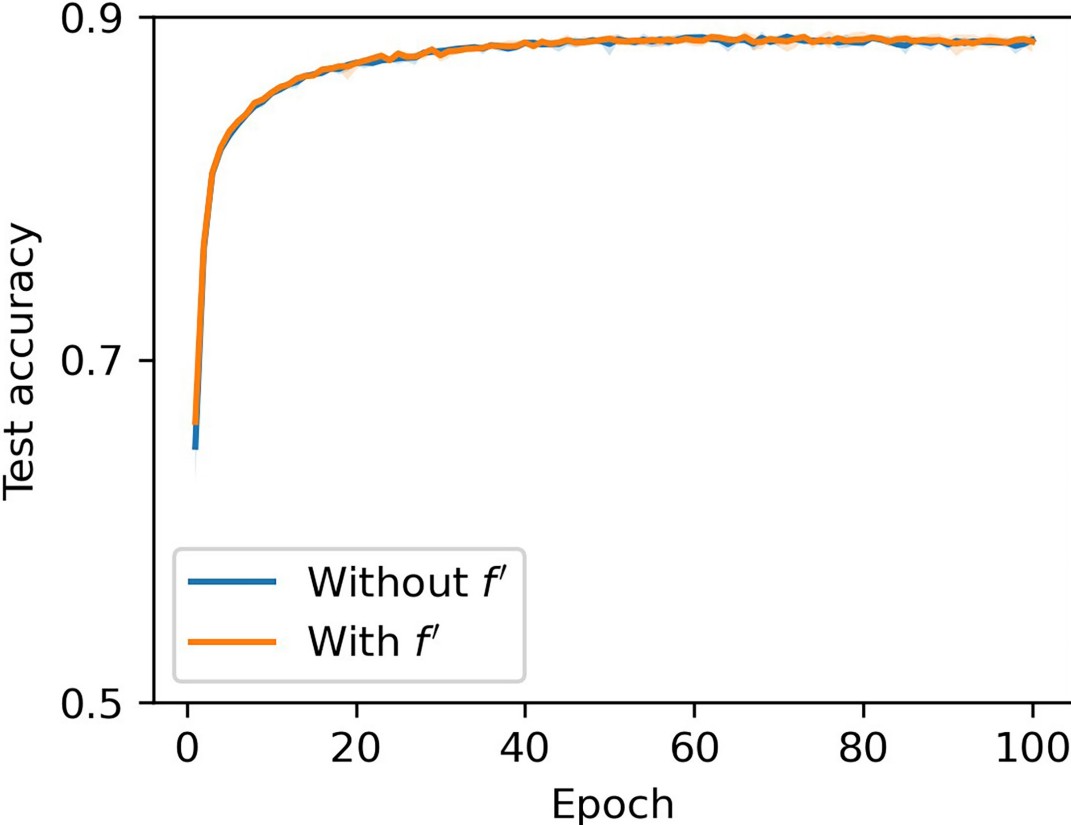

**Appendix 1—figure 1.** Test accuracies on FMNIST with $f'$ and without $f'$. A 2-hidden-layer model by default setting is used.

The online version of this article includes the following source data for appendix 1—figure 1:

**Appendix 1—figure 1—source data 1.** The source data for *Appendix 1—figure 1*.

# Appendix 2

## Two-alternative forced choice (2AFC) task

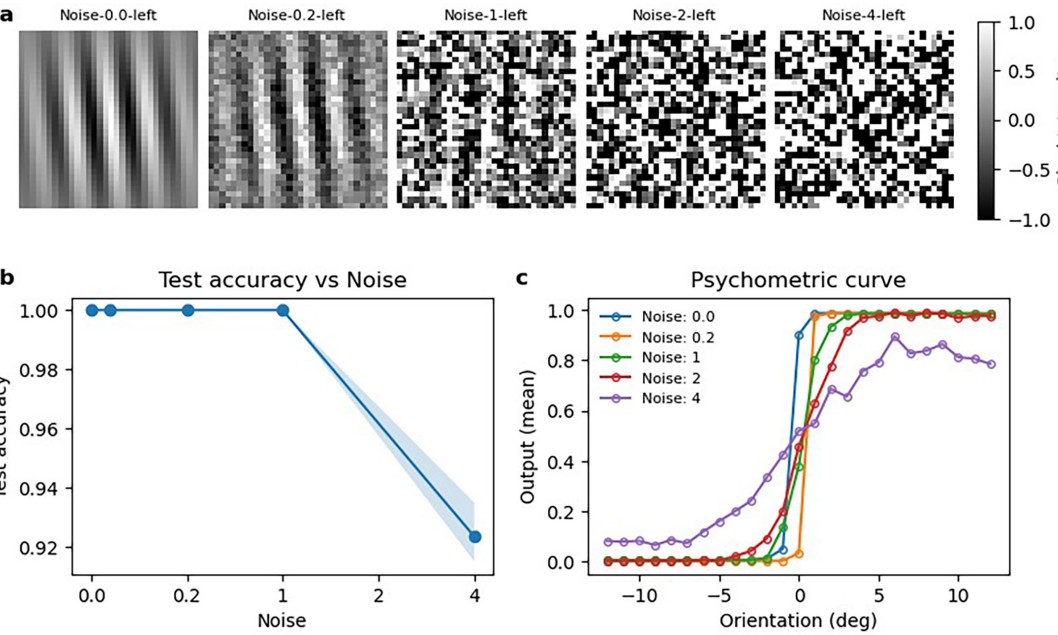

**Appendix 2—figure 1.** 2AFC task. (**a**) Sample training images tilted 6 degrees to the left with different noise levels. (**b**) Test accuracy versus noise level. (**c**) Psychometric Curve. Noise level is defined as the standard deviation of the normal (Gaussian) distribution noise.

The online version of this article includes the following source data for appendix 2—figure 1:

**Appendix 2—figure 1—source data 1.** The source data for *Appendix 2—figure 1*.

 2AFC (Two-Alternative Forced Choice) is a psychophysical task where a subject or a model must make a choice between two categories of stimuli, with a baseline accuracy of 50% for random guessing. We have generated a training (test) dataset consisting of 1000 (400) images of dark and light bands tilted by 6 degrees to the left or right with different noise levels (*Appendix 2—figure 1a*). We used a two-hidden-layer model with 256 neurons in each hidden layer for this study and trained it for 20 epochs with Adam optimizer and 2e-2 learning rate. The output of the model between 0 and 1 maps the model's confidence in left or right tilting. 0 predicts 100% left, and 1 100% right. *Appendix 2—figure 1b* shows that our model achieves 100% accuracy with noise level less than 1. For images with higher level noise, the test accuracy decreases. Then we use samples of orientation from –12 degrees to 12 degrees to stimulate the model. By recording the output under different stimulus, a psychometric curve (usually S-shaped) that characterizes perceptual sensitivity and discriminative threshold can be plotted (*Appendix 2—figure 1c*; *Gold and Ding, 2013*; *Cheng et al., 2025*).

## Appendix 3

### The balance between error feedback scaling $\beta$ and spectral radius

The amplitude of the error (gradient) signal is affected by the feedback coefficient $\beta$ in *Equation 1* in the main test and by the spectral radius of modular RNN. In the main text, $\beta$ is fixed to 1 in the learning phase. If $\beta$ is allowed to vary, the spectral radius should also vary to balance the influence of $\beta$ on error signal (*Appendix 3—figure 1a,b and c*). When the number of layers increases, only a small range of combinations yields good performance (*Appendix 3—figure 1c*). We further show the amplitude of the error with different $\beta$ and SR in *Appendix 3—figures 2 and 3*. Unsurprisingly, smaller $\beta$ (or SR) leads to gradient vanishing, larger $\beta$ (or SR) leads to gradient explosion (or non-convergence of RNN). There exists an optimal $\beta$ (or SR) for a task. These results suggest that RNNs in AP can provide additional means to regulate the amplitude of error signal, so that deeper models could be trained.

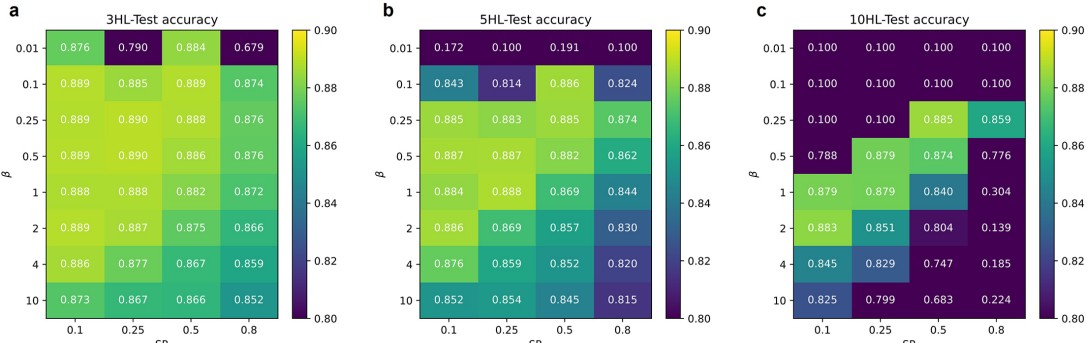

**Appendix 3—figure 1.** The test accuracy of 3-hidden-layer (**a**), 5-hidden-layer (**b**), and 10-hidden-layer (**c**) with varying $\beta$ and spectral radius. The models were trained for 100 epochs. Each experiment is repeated twice.

The online version of this article includes the following source data for appendix 3—figure 1:

**Appendix 3—figure 1—source data 1.** The source data for *Appendix 3—figure 1*.

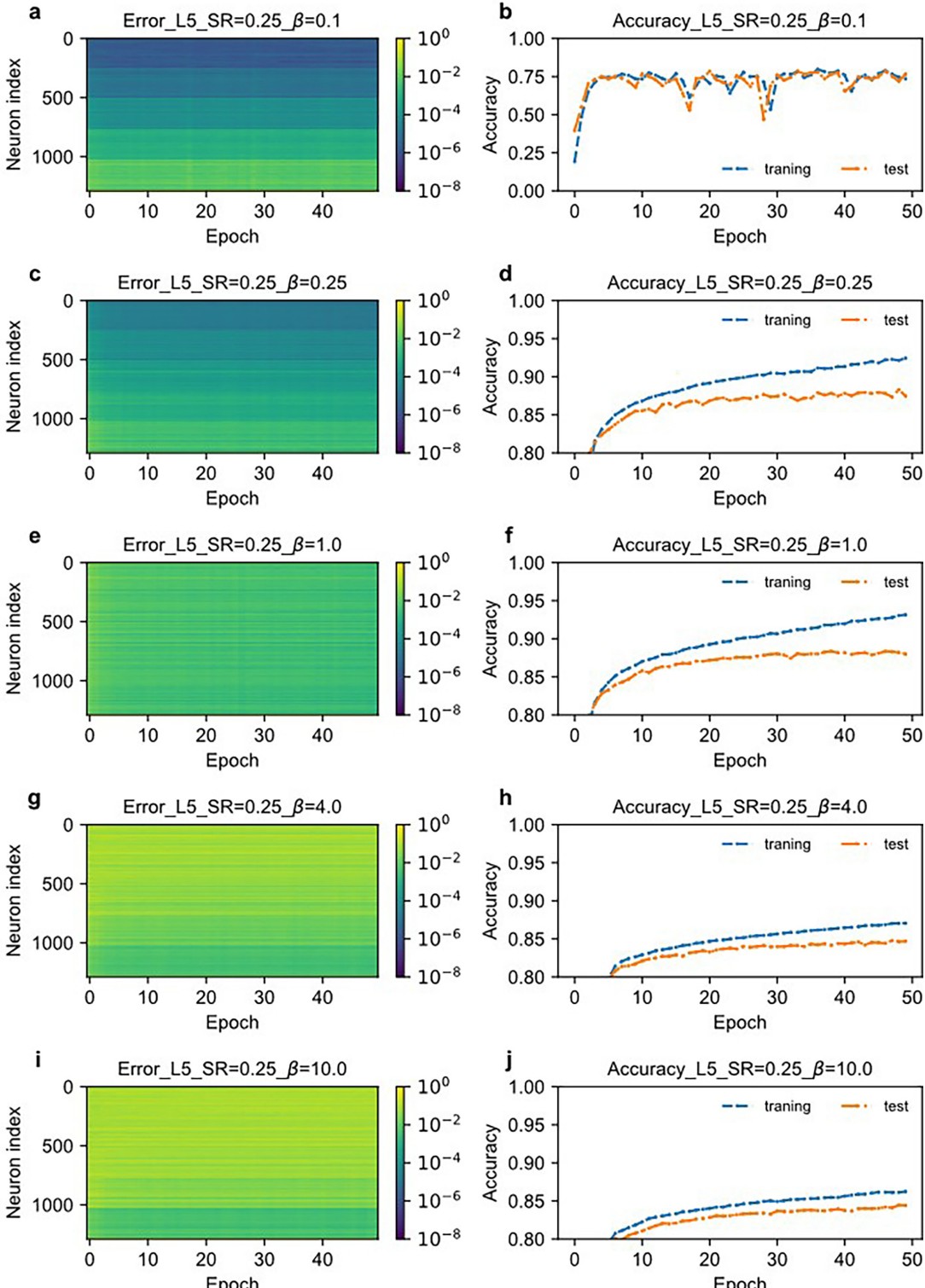

**Appendix 3—figure 2.** Mean absolute values of error of each SI neuron and accuracy versus epochs at different $\beta$. The error values are collected from the last batch of training of a five-hidden-layer RNN. All SI neurons in the hidden layers and the output layer are indexed from the input to the output layer, i.e., neurons with smaller index are closer to the input layer. In a, small-indexed neurons exhibit weak error signals, which manifests gradient vanishing. With increasing $\beta$ (**a, c, e**), these error signals become larger, leading to better performances (see **b, d, f**). But excessively larger $\beta$ causes gradient explosion (see **e, g, i**), ultimately reducing testing accuracies (**f, h, j**). The network has been trained for 50 epochs in each case.

The online version of this article includes the following source data for appendix 3—figure 2:

**Appendix 3—figure 2—source data 1.** The source data for *Appendix 3—figure 2*.

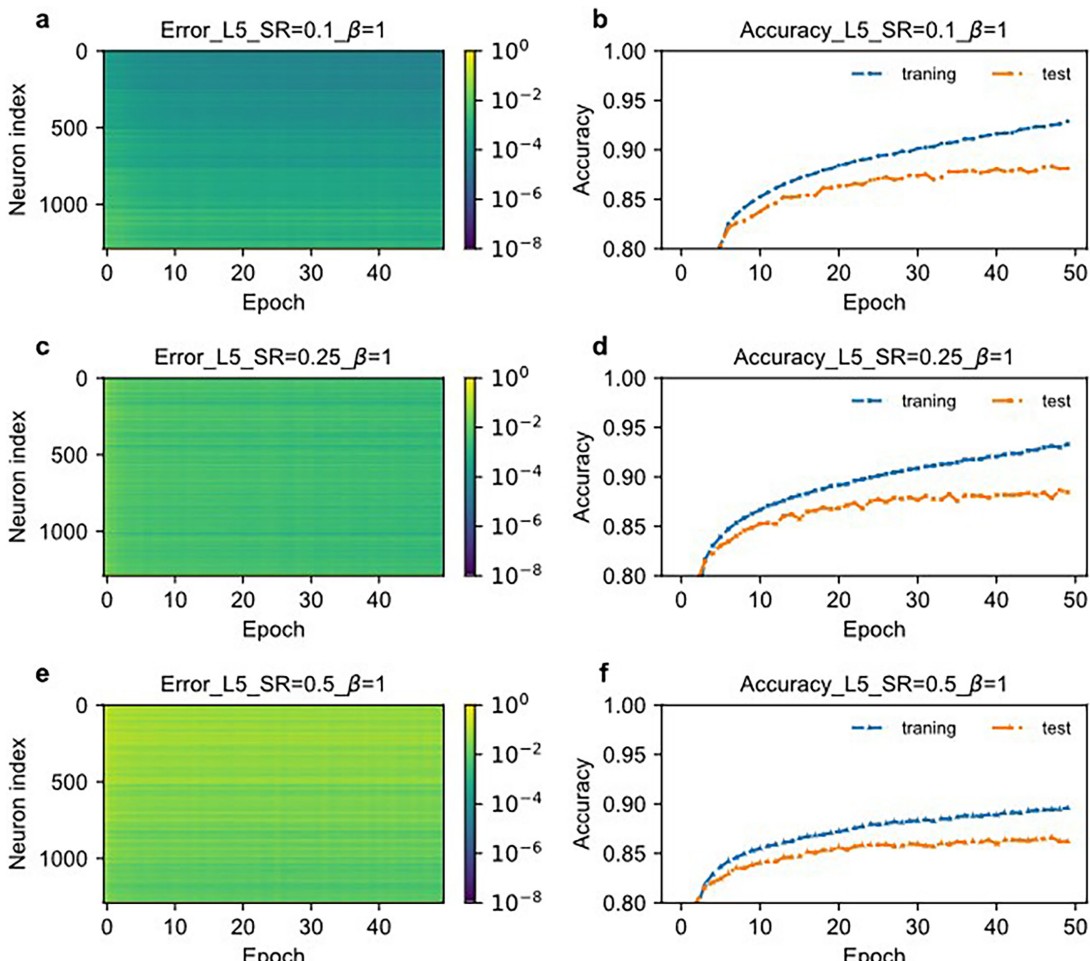

**Appendix 3—figure 3.** Mean absolute values of error of each SI neuron and accuracies versus epochs at different SR. The error values are collected from the last batch of training a five-hidden-layer RNN. All SI neurons in the hidden layers and the output layer are indexed from the input to the output layer, i.e., neurons with smaller index are closer to the input layer. The effect of increasing SR is similar to that of increasing $\beta$ as observed from *Appendix 3—figure 2*.

The online version of this article includes the following source data for appendix 3—figure 3:

**Appendix 3—figure 3—source data 1.** The source data for *Appendix 3—figure 3*.

## Appendix 4

### Why can two tasks be learned simultaneously?

Here, we provide further discussion on why one network can learn two tasks as shown in *Figure 5* in the main text.

### Case of alternating configuration

In this case, we configure the network for one task and train it for one epoch of data, then we switch to another task, so on and so forth (Algorithm 2). Let us denote the network's learnable parameters at $i$th step with $\theta_i$, the weight update for one task can be described by:

$$\theta_{t+1} = \theta_t - \eta \frac{\partial \text{Loss}_j\left(\theta_t\right)}{\partial \theta_t} = \theta_t - \eta \nabla \text{Loss}_j\left(\theta_t\right), \tag{A5}$$

where $\eta$ is the learning rate and $\text{Loss}_j$ is the loss of task-$j$. In one epoch, the weights are updated $n_{batch}$ times for one task. Here, we consider it as one update. Consider two consecutive updates, one for each task:

$$\theta_t = \theta_{t-1} - \eta \nabla \text{Loss}_1\left(\theta_{t-1}\right), \tag{A6}$$

$$\theta_{t+1} = \theta_t - \eta \nabla \text{Loss}_2\left(\theta_t\right). \tag{A7}$$

We assume that the loss functions are smooth functions with Lipschitz continuous gradients, i.e., there exists a constant $L$ such that $\left\|\nabla \text{Loss}_j\left(\theta_t\right) - \nabla \text{Loss}_j\left(\theta_{t'}\right)\right\| \leq L \left\|\theta_t - \theta_{t'}\right\|$. We can use Taylor expansion and ignore higher order terms to approximate the $\nabla \text{Loss}_2\left(\theta_t\right)$:

$$\nabla \text{Loss}_2\left(\theta_t\right) = \nabla \text{Loss}_2\left(\theta_{t-1} - \eta \nabla \text{Loss}_1\left(\theta_{t-1}\right)\right) \approx \nabla \text{Loss}_2\left(\theta_{t-1}\right) - \eta \nabla^2 \text{Loss}_2\left(\theta_{t-1}\right) \nabla \text{Loss}_1\left(\theta_{t-1}\right), \tag{A8}$$

Substitute the *Equation A6* and *Equation A8* into *Equation A7*:

$$\theta_{t+1} \approx \theta_{t-1} - \eta \left(\nabla \text{Loss}_1\left(\theta_{t-1}\right) + \nabla \text{Loss}_2\left(\theta_{t-1}\right)\right) + \eta^2 \nabla^2 \text{Loss}_2\left(\theta_{t-1}\right) \nabla \text{Loss}_1\left(\theta_{t-1}\right). \tag{A9}$$

Due to the small learning rate, the high-order term can be omitted:

$$\theta_{t+1} \approx \theta_{t-1} - \eta \left(\nabla \text{Loss}_1\left(\theta_{t-1}\right) + \nabla \text{Loss}_2\left(\theta_{t-1}\right)\right). \tag{A10}$$

Therefore, the net update direction can reduce the losses of both tasks, and one network can learn two tasks with alternating configurations.

### Case of simultaneous input

If we input samples of two tasks simultaneously as described in Algorithm 3, the feature information and the error information will interfere with each other. The update rule becomes:

$$\theta_{t+1} = \theta_t - \eta \cdot \nabla g\left(\text{Loss}_1\left(\theta_t\right), \text{Loss}_2\left(\theta_t\right)\right), \tag{A11}$$

here, $g$ denotes the composite error function assumed to be monotonically dependent on $\text{Loss}_1\left(\theta_t\right)$ and $\text{Loss}_2\left(\theta_t\right)$. Due to the averaging effect of batch learning, hypothetically the mean gradient $M\left[\nabla g\left(\text{Loss}_1, \text{Loss}_2\right)\right]$ of composite loss with simultaneous input can be approximated by the sum of individual gradients of the two tasks $\nabla \text{Loss}_1 + \nabla \text{Loss}_2$. The results show that the network can simultaneously learn both tasks despite the interferences, thus supporting the hypothesis. The uncorrelatedness of tasks ensures that the interferences on gradient are smeared out in batch averaging and time averaging, which stabilizes the learning process.

To further corroborate the hypothesis above, we further examine the batch cosine similarity of error vectors (weight adjustments) between the two ways of two-task training shown in *Figure 5*. We calculated the batch cosine similarity (a) and the batch cosine similarity of mean weight updates (b) for each batch. $e1 - e5$ denote the errors of RNNs L1-4 and the output error of the Task 1. *Appendix 4—figure 1* shows that the cosine similarity of $e2$ and $e5$ is exactly 1, because these errors are solely determined by task 1 (*Figure 5b*). The cosine similarity of $e1, e3$ and $e4$ fluctuates, since the

errors of shared RNNs are susceptible to interference when the two tasks are trained at the same time. Nevertheless, the batch cosine similarities are remarkably close to 1, and the batch cosine similarities of weight updates are even more so (**Appendix 4—figure 1b**), thus further corroborating the hypothesis above. Consistent with this observation, analogous conclusions can be drawn from **Appendix 4—figure 2**, which corresponds to the pathway presented in **Figure 5c**.

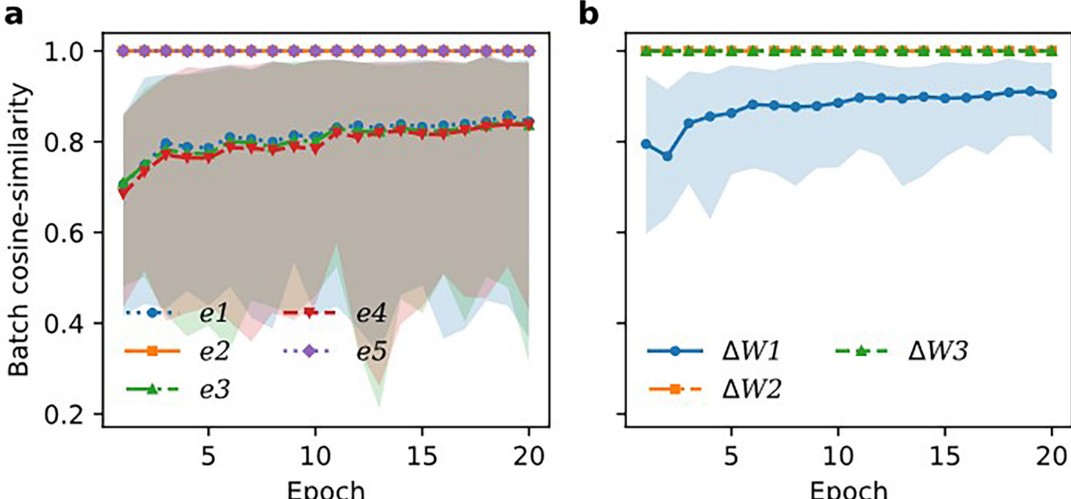

**Appendix 4—figure 1.** The batch cosine similarity of error (**a**) and weight updates (**b**) of task 1, computed with and without simultaneous input of two tasks in **Figure 5b**. Training consisted of 20 epochs with 120 batches per epoch (500 samples per batch). $\Delta W1 - \Delta W3$ corresponds to the update of weights $x_1 - \mathrm{L}(1)$, $\mathrm{L}(1) - \mathrm{L}(2)$ and $\mathrm{L}(2) - y_1$.

The online version of this article includes the following source data for appendix 4—figure 1:

**Appendix 4—figure 1—source data 1.** The source data for **Appendix 4—figure 1**.

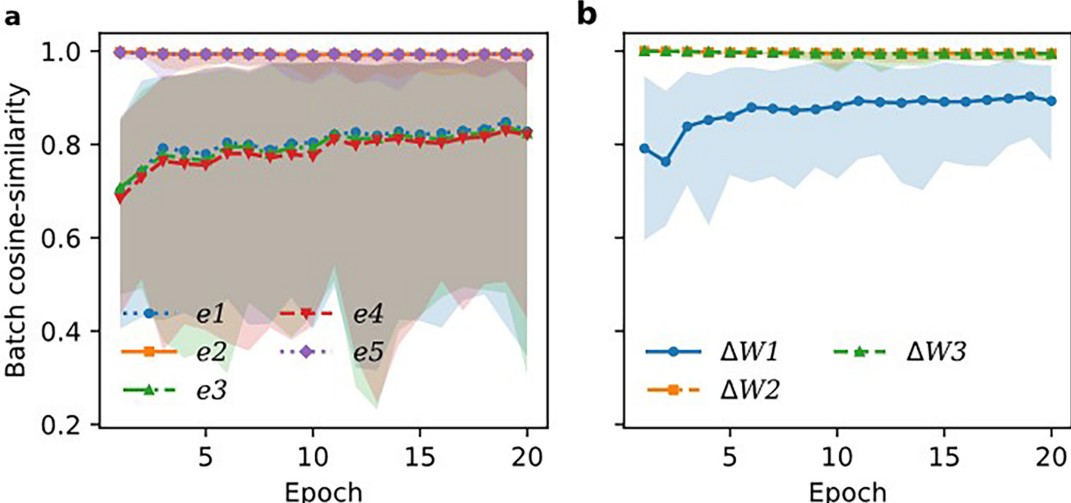

**Appendix 4—figure 2.** The batch cosine similarity of error (**a**) and weight updates (**b**) of task 1, computed with and without simultaneous input of two tasks in **Figure 5c**. Training consisted of 20 epochs with 120 batches per epoch (500 samples per batch).

The online version of this article includes the following source data for appendix 4—figure 2:

**Appendix 4—figure 2—source data 1.** The source data for **Appendix 4—figure 2**.

