## [Editor Report]

This article addresses a particular problem of current theories of supervised learning in cortical hierarchies: the fact that the inference signal afferents are not targeted reciprocally by potentially error-related connections. The authors propose an important idea where such a mismatch can be bridged by recurrent connections. The theory is supported by solid theoretical simulations.

---

## [Decision Letter]

**Decision letter after peer review:**

Thank you for submitting your article "Adjoint propagation of error signal through modular recurrent neural networks for biologically plausible learning" for consideration by *eLife*. Your article has been reviewed by 3 peer reviewers, one of whom is a member of our Board of Reviewing Editors, and the evaluation has been overseen by a Reviewing Editor and Michael Frank as the Senior Editor. The reviewers have opted to remain anonymous.

Essential Revisions:

1) Revise main text and figure for biology audience.

2) Improve comparisons with other models.

It would be good to add a discussion on temporal tasks and spiking networks (mentioned by the reviewers), but these are deemed outside the scope of the current article.

*Reviewer #1 (Recommendations for the authors):*

The authors have a explored a beautiful idea that is making a lot of sense from a biological point of view. I thought the article made a number of creative and potent simulations which would benefit from a wide neuroscience readership. The inclusion of the spectral radius as a parameter was a careful choice that is increasing the scope of their article. Also, the study of gated FMNIST+MNIST networks is also a conceptually relevant direction.

There are a number of significant weaknesses. My comments here are mainly trying to help the authors in improving their manuscript to get a wider readership. The main issues are in my opinion the readability of the main text and main figures, which I found fairly difficult and therefore I imagine it will gate any non-computational neuroscientist out. There were a number of places where assumptions were presented as facts, which I think it both unnecessary and harmful. Similarly, I was hoping that the authors would make a more square statement about the apparent limits of the algorithm in terms of performance and scaling, not including this statement is also unnecessary and harmful: yes, if you are submitting to NeurIPS performance is all that matters, but in biology it is different, a lower performance at something often means a better performance at something else, so it is not perceived as a problem per se. Finally, I thought that the results would be better served with an inhouse comparison and an assessment of the effects of ongoing dynamics in different parts of the adjoint network.

General comment. The manuscript is very heavy in acronyms. The mathematical notation is difficult to parse with a number of IMO unnecessary subscript and superscripts.

The article does a poor job at talking to non-computational scientists. The core architecture and the core argument is done with a lot of computational jargon, a figure that is loaded with acronyms and difficult to parse, and text that relies more on the math than describing in words what is being modeled. It would be important to improve the language. Figure b is at odds with the main text 'each functional block accounts for a quarter of the total neurons', since there are neurons represented that do not belong to either SI RE RI or SE.

Figure 1. I don't understand what is being conveyed here. I hoped for something that would make the architecture clear as the text is tedious in that respect, but I struggle with this figure. I don't know what a is supposed to mean other than the trivial 'there connections in the brain'. Where in the brain do the authors think their 4-part unit takes place?

Line 71 "This assumption is supported by biological evidence of specialized neurons or the apical dendrites of pyramidal neurons [37-40], and widely adopted by other learning frameworks, such as target propagation [11,41,42] and local representation alignment [15,43]." Better nuances are needed. That some adopt it is not enough to say it is widely adopted. There are many works that attempt to avoid this unbiological aspect (e.g. Greedy… Costa, or Payeur… Naud). And there is no biological evidence for two phases: the papers cited use biological mechanism for simulating two phases, but they don't show two phases in neuronal networks recorded experimentally, these phases are possible and therefore framed as assumptions in NeuroAI theories. It would be important for the validity of claims of this paper that the idea of two phases be presented as such. Relatedly, the phase dependence as well as the particular signs of the Hebbian/anti-Hebbian terms has not been seen in experiments (Eq. 2). It does not mean it does not exist, but it is an assumption that should be described as such rather than relying on its likeness to Hebbian and contrastive learning principles.

Line 124. It would be very helpful to provide an in-house MNIST accuracy for the networks without the sender/receiver constraint. This accuracy does not compare well with SOTA, but it's clearly not the goal here; hence the comparison with the relevant point is important. Readers are used to MNIST being taken as a community-benchmark and a toy task, but there is the uncanny tendency for reader to revert to the idea that it's a benchmarking task. And it's not what is being used for here. Another approach would be to steer away from the community benchmark and use a neuro task.

It is very nice that the authors have thought of the spectral radius as an important parameter regulating the efficacy of their method.

Ongoing activity. It would seem that the main problem with having error sent at other locations than inference signals is not so much the connectivity itself, as one could indeed think of a relay cell scenario. But the main problem would be if the different cells are engaged in different tasks. It would seem that the authors think that the algorithm would also be robust to that but in my understanding, this has not been shown. The simulations in Figure 5 get close to this, but this would require to explicitly have some of the inference input getting to the error neurons. The article is lacking precision as to how Figure 5a is conducted (how are the input paths chosen? All part of the 'RI' parts?).

Lack of clarity relating to Figure 4. Line 190, I could not comprehend this section , starting with " We illustrate two concepts of error… (STE)" What is a 'concept of error representation'? They seem to be different type of presentations/dynamics of the error signal, but the terminology 'concept of error representation' seem to indicate something more than that. What are sub-trajectories? Where do these come from? What's being plotted in 3b and 3c? What's X-Y-Z here?

Rework Figure 5. The arrangement of Figure 5 is in my opinion unnecessarily difficult to parse. A has one network repeated twice, and b has one network twice as large. These are conceptually very different and the visual symmetry is deceptive. In addition, Figure 5d. It would seem that the curves are superposed but we do not see where.

The discussion was missing a statement about the role of the spectral radius in network. It has been argued as being potentially large, but also as potentially small. So the relationship with performance can imply a trade-off… or not.

A small note, I thought the paper was not very precise as to the problem of lack of reciprocity in cortical networks. I know the theory of Payeur et al., Naud (2021; NN). is based on mainly the thick tufted, extra telencephalic (ET) L5 pyramidal cells, which are only part of the network L5 network, which makes the question a little more precise, but some precision would be beneficial for the Sacramento et al. Senn NeurIPS Theory as well. These L5-ET cells are thought to be the main part of the L5 network that changes the stimulus representation (Moberg et al. Takahashi 2025, BioRxiv), and these cells are not those receiving reciprocal inputs from higher order inputs (Young, et al. Petreanu 2021, *eLife*). The other set of cells, the IT cells, seem to be engaged in a reciprocal network between the L5 of the upper layers. The cells also receive feedforward inputs, but they seem less to undergo learning (again the Moberg paper). It would seem to me that this could correspond to an adjoint network where all cells receive the inference signal but only a portion receive the teaching signal.

*Reviewer #2 (Recommendations for the authors):*

The work in this paper should be seen as a stepping stone to something larger. The research space of biologically plausible credit assignment is relatively small (and overall rather under-explored), and this work can be used as a grounding / foundation for future ideas. The authors' demonstration of their methodology's ability to not only change the number of error propagating/receiving neurons (within a multi-region recurrent neural net, i.e., MR-RNN) but also its ability to reuse (neuronal) structures is a nice step forward for computational neuroscience / neuroAI. Beyond the reuse of structures for different tasks, the ability to piece ("compose") together different structures to accomplish the same task shows robustness in the approach's / framework's design.

Without detracting from the work accomplished by the authors, there still are a few things that need to be kept in mind when reading/viewing this work. Specifically the base line problems / benchmarks used, i.e., MNIST and FMNIST, are the most basic of baselines / benchmarks. This means that there is no guarantee that this framework/methodology will scale to larger (data)sets such as CIFAR, ImageNet, or other larger natural images (or other high dimensional, more complex collections of data patterns).

Please find below a few things that the authors should take care of:

1) Small edit on line 362 – there is a typo of "is mall" that I believe should be "is small".

2) All figures could benefit from larger font size/plots; please adjust accordingly to make things more presentable/viewable to readers

3) Things to address in the manuscript:

a) The authors need to comment on why feedback alignment and adjoint propagation lose accuracy as the model size increase while back propagation gains accuracy (as seen in later results)? What happens if the size of the model increases dramatically, is the accuracy adversely affected? A good of writing should be provided in the main text to address troubling issue and, ideally, a few extra experiments that examine the effect of increasing model size/capacity (this would help to highlight that perhaps, while bio-plausible methods address some limitations of backprop-based DNNs, they might come at the cost of some parameter inefficiency or greater model architecture/parameter complexity).

b) As of right now this work deals with entirely continuous values, with the RNN being the analog for biological recurrence. Have the authors explored or considered other stateful biological models such as spiking neural networks (SNNs)? If so, how did SNNs preform as compared to the tested model? This is important to address (in writing and, ideally, experimentally) as there is mention of hardware implementations in the Discussion section, since energy-efficiency tends to emerge as a result of these specific styles of networks (in terms of structure), not just from sparsity. At the very least, some consideration of spiking-driven dynamics should be provided given that continuous RNN models typically look like rate-coded approximations of SNNs (and some SNNs, such as those based on leaky integrate-and-fire cells, follow similar neuronal dynamics/flow).

*Reviewer #3 (Recommendations for the authors):*

This study presented a novel neural network architecture for solving supervised classification tasks, aiming at achieving high biological plausibility. Traditionally, such tasks were solved by multi-layered feed-forward network trained according to the backpropagation (BP) algorithm, i.e., trained by error feedback with the weights that are transposed copies of the (temporally evolving) forward weights (called the symmetric feedback); this was argued to be biologically implausible. More recently, it was shown that comparable performance could be obtained even if the symmetric feedback is replaced with fixed random feedback, which is biologically more plausible, because the forward weights become aligned to the feedback weights (called the feedback alignment (FA)). This study examined an architecture (named the Adjoint Propagation (AP)) deviated from the feed-forward network, while using FA. Specifically, whereas the feed-forward network consists of sequentially connected layers, each of which consists of neurons without mutual connections, AP consists of sequentially connected modules, each of which consists of neurons with mutual connections that are learnable (i.e., recurrent neural network (RNN)). Each module consists of four functional blocks: receiving inference signal (RI), sending inference signal (SI), receiving error (RE), and sending error (SE), and interaction across modules (and also between the input and output) are selectively done according to these prespecified blocks. The authors derived update rules for across-module and within-module weights according to a gradient descent, with approximations. Different from BP and in the same manner as FA, the error feedback weights were fixed, and forward weights aligned to them. So in this regard, AP is biologically more plausible than BP and has the same level of plausibility as FA. However, the module of AP has recurrent connections whereas the layer of the feedforward network (used in FA) does not, and in this regard AP can be biologically more plausible than FA, as the authors argue (but see my comment on this below). As for performance in the examined two tasks (using the public data set), AP is largely comparable to (but can be slightly inferior to) FA and BP.

Strengths:

1) The proposed architecture, sequentially connected modules, each of which has functional blocks and learnable recurrent connections, looks indeed closer to a network of multiple cortical areas than the conventional feed-forward network. Although the assumed structure is somewhat artificial (four prespecified functional blocks) and not exactly the same as the actual structure of cortical area, the proposed model can be a good model of cortical information processing.

2) Error representation by a difference between two trajectories of neural states in the recurrent module is an interesting idea, and because it can be calculated far before the iteration of weight update becomes converged, it can in principle speed up learning.

3) Reuse of the same network for multiple different tasks is an important thing to examine, and the result looks potentially interesting.

Weakness:

1) Whether the representation of error by a difference between two trajectories of neural states can be implemented in a biologically plausible manner seems unclear, as the authors mentioned in Line 198-200. Because the primary focus of this study is biological plausibility, this point is a severe limitation.

2) In terms of performance (Table 1), although AP can be said to be largely comparable to FA and BP, precisely speaking, AP is slightly inferior to FA, which is slightly inferior to BP. It is unclear this slight difference in the performance could be more prominent in other tasks (e.g., those for which FA was shown to perform worse than BP).

3) As for the reuse of the same network for multiple different tasks, comparison with other models is lacking, and it is difficult to judge how good the shown reusability of AP is.

4) A prominent advantage of recurrent neural network, in general, is that it can represent and learn a temporal sequence of information such as language. But this study examined static classification tasks (MNIST dataset: handwritten digits, and FMNIST dataset: grayscale images of 10 clothing categories), and whether the proposed AP can also learn the temporal sequence appears to be unclear.

5) In the Abstract and Introduction, the authors pointed out the biological implausibility of BP, and introduced AP as a more plausible architecture, while FA was not cited until the middle of the Results. But because FA is an established model that is more plausible than BP and moreover the authors' AP actually also uses FA as a learning principle, it should be fairer to introduce FA in the Abstract/Introduction and framed AP in comparison with FA rather than (or in addition to) BP.

6) In the originally introduced setting of AP, separate phases for inference and learning are required, and its biological plausibility is unclear. The same problem also exists for BP and FA, but AP requires learning of recurrent network, which usually requires a certain number of iterations, and so the plausibility problem can be said to be severer for AP with this setting than BP or FA. The error representation by a difference between trajectories can beautifully resolve this issue, but it has plausibility issue on its own as I commented in the public review – weakness 4).

7) In Figure 3c, cases with more iterations appear to have lower accuracy. Why is it?

8) In Figure 4d, even if the number of RE neurons was set to fairly small (and the network became closer to reservoir computing as the authors mentioned), the performance remained largely comparable. The authors' description sounds it can be a good feature, but doesn't it indicate that the merit of learnability of recurrent connections (as opposed to reservoir computing) is rather limited (even though the performance was better than EP and so there was at least some merit)? And if so, is the reason for it the approximation (Line 369-371) made in the derivation of update rule for recurrent connections, in that this approximation is similar to the omission of non-local terms done in RFLO (Murray, 2019, *eLife*) or e-prop (Bellec et al., 2020 Nat Commun), which was shown (in the case of RFLO) to degrade the performance even more than random feedback (compared with symmetric feedback) did (Figure 2d of Murray, 2019)?

9) The authors used the term "mixed selectivity" to refer to selectivity across different tasks. Does this match its original terminology (doesn't it refer to selectivity for multiple features within a task)?

---

## [Author Response]

Essential Revisions:Reviewer #1 (Recommendations for the authors):The authors have a explored a beautiful idea that is making a lot of sense from a biological point of view. I thought the article made a number of creative and potent simulations which would benefit from a wide neuroscience readership. The inclusion of the spectral radius as a parameter was a careful choice that is increasing the scope of their article. Also, the study of gated FMNIST+MNIST networks is also a conceptually relevant direction.

R1-1: We thank the reviewer for the positive comments on our idea.

There are a number of significant weaknesses. My comments here are mainly trying to help the authors in improving their manuscript to get a wider readership. The main issues are in my opinion the readability of the main text and main figures, which I found fairly difficult and therefore I imagine it will gate any non-computational neuroscientist out. There were a number of places where assumptions were presented as facts, which I think it both unnecessary and harmful.

R1-2: We have improved the manuscript's readability by revising the text and figures. The revisions include a clearer presentation of key assumptions—such as temporal difference computation and Hebbian/anti-Hebbian terms—and a more detailed exposition of the role constraint and the two-phase learning process. See also R1-5, R1-6, R1-7 and R1-8 below.

Similarly, I was hoping that the authors would make a more square statement about the apparent limits of the algorithm in terms of performance and scaling, not including this statement is also unnecessary and harmful: yes, if you are submitting to NeurIPS performance is all that matters, but in biology it is different, a lower performance at something often means a better performance at something else, so it is not perceived as a problem per se.

R1-3: We fully agree that discussing the limits of the algorithm in terms of performance and scaling is important. We have added discussions about such limitations in AP from line 240 to line 250 and from line 256 to line 258 in the revision.

Briefly, our AP algorithm is scalable to deeper networks (see more detailed reply at R2-5 below).

Finally, I thought that the results would be better served with an inhouse comparison and an assessment of the effects of ongoing dynamics in different parts of the adjoint network.

R1-4: We appreciate this valuable advice. We have performed additional experiments in which we train the network on two tasks simultaneously. The results show that the network can learn two tasks at the same time (Figure 5b-c, Figure 5e-f). While the inference/error signals of the two datasets do influence each other, this has a negligible impact on performance because of the time averaging of gradient over multiple batches and epochs (more discussion in Appendix 4).

General comment. The manuscript is very heavy in acronyms. The mathematical notation is difficult to parse with a number of IMO unnecessary subscript and superscripts.The article does a poor job at talking to non-computational scientists. The core architecture and the core argument is done with a lot of computational jargon, a figure that is loaded with acronyms and difficult to parse, and text that relies more on the math than describing in words what is being modeled. It would be important to improve the language. Figure b is at odds with the main text 'each functional block accounts for a quarter of the total neurons', since there are neurons represented that do not belong to either SI RE RI or SE.

R1-5: We have made several improvements, including the removal of the SI, RE, RI, and SE subscripts. By consulting similar works, we have also minimized the use of mathematical symbols while maintaining scientific rigor.

Figure 1b (Figure 1c in revision) illustrates a more general scenario. That 'each functional block accounts for a quarter of the total neurons' is the default setting for initial demonstrations. The neurons in an RNN do not necessarily take one of the roles of SI, RE, RI or SE. We have clarified this point in the main text (line 78).

Figure 1. I don't understand what is being conveyed here. I hoped for something that would make the architecture clear as the text is tedious in that respect, but I struggle with this figure. I don't know what a is supposed to mean other than the trivial 'there connections in the brain'. Where in the brain do the authors think their 4-part unit takes place?

R1-6: Figure 1a depicts the macro-scale recurrent connections across different areas of brain, e.g., between sensory cortex and thalamus. There are also recurrent connections among different cortical areas and among different layers (Figure 1b in the revision). Figure 1c (Figure 1b in the initial submission) generalizes an abstract model of multi-region recurrent neural network modules from these observations in neuroanatomy.

The name of the four blocks, SI, RI, SE, RE, only discriminate the roles of neurons in certain computational task, and does not require them to be fundamentally different. We make this point clear in the revision in line 79.

Line 71 "This assumption is supported by biological evidence of specialized neurons or the apical dendrites of pyramidal neurons [37-40], and widely adopted by other learning frameworks, such as target propagation [11,41,42] and local representation alignment [15,43]." Better nuances are needed. That some adopt it is not enough to say it is widely adopted. There are many works that attempt to avoid this unbiological aspect (e.g. Greedy… Costa, or Payeur… Naud).

R1-7: We thank the reviewer for pointing out this important nuance about SE neuron assumption. The text is now revised (line 82) to better represent the status of this assumption. Whether or not the computing of temporal difference is dispensable for the brain is presently unsure. However, this assumption is supported by biological evidence of specialized neurons, the apical dendrites of pyramidal neurons and local circuit motifs [Silberberg and Markram, 2007, Neuron 53: 735-746; Kubota, 2014, Curr Opin Neurobiol 26: 7-14; Leinweber et al., 2017, Neuron 95: 1420-1432; Roelfsema and Holtmaat, 2018, Nature Reviews Neuroscience 19: 166-180; Sacramento et al., 2018, Advances in Neural Information Processing Systems 31]. Therefore, similar assumption is adopted by other learning frameworks, such as target propagation.

And there is no biological evidence for two phases: the papers cited use biological mechanism for simulating two phases, but they don't show two phases in neuronal networks recorded experimentally, these phases are possible and therefore framed as assumptions in NeuroAI theories. It would be important for the validity of claims of this paper that the idea of two phases be presented as such. Relatedly, the phase dependence as well as the particular signs of the Hebbian/anti-Hebbian terms has not been seen in experiments (Eq. 2). It does not mean it does not exist, but it is an assumption that should be described as such rather than relying on its likeness to Hebbian and contrastive learning principles.

R1-8: We agree that the assumption of two distinct phases should not be presented as biologically established.

We would like to point out a semantic nuance of the term ‘two phases’. In our opinion, the learning process in biological neural network is two-phase (two-stage) by nature. The network has to be stimulated by an input (from sensors or internal memory) first and reach a prepared state. Then the label is presented to the network, and the network adapts itself to correlate the input and its label. That is to say, the stimulus has to be ‘online’ during training [Payeur et al., 2021, Nature Neuroscience 24: 1010-1019]. In our AP framework, the sensory stimulation can be kept ‘online’, the term “two phases” simply distinguishes whether the label is presented to the network. The inference and learning proceed in a quasi-continuous manner. We have enhanced our elaboration from line 90 to line 92 and line 204 to line 212.

The AP framework supports both types of two-phase learning, as illustrate in Figure 3.

We now clearly present Hebbian/anti-Hebbian terms as an assumption. Admittedly, experimental observations are lacking even though the two terms are related to LTP and LTD of synapses (line 128 to line 130) [Hyman et al., 2004, The Journal of neuroscience : the official journal of the Society for Neuroscience 23: 11725-11731].

Line 124. It would be very helpful to provide an in-house MNIST accuracy for the networks without the sender/receiver constraint.

R1-9: We appreciate the reviewer's suggestions. The in-house MNIST accuracy for the networks without the sender/receiver constraint is given in Figure 4e-f. AP without constraint performs better in shallow structures, but fails with 10-hidden-layer. The sender/receiver constraint allows to regulate gradient amplitude of deeper networks by spectral radius (more evidence in Appendix 3).

This accuracy does not compare well with SOTA, but it's clearly not the goal here; hence the comparison with the relevant point is important. Readers are used to MNIST being taken as a community-benchmark and a toy task, but there is the uncanny tendency for reader to revert to the idea that it's a benchmarking task. And it's not what is being used for here. Another approach would be to steer away from the community benchmark and use a neuro task.

R1-10: Indeed, improving accuracy is not the goal. We thank the reviewer for pointing out the risk of misleading readers. We have clarified the goal in the text (line 142).

We have also carried out a neuro task Two-Alternative Forced Choice (2AFC), which works well as expected. Neuro tasks like 2AFC are less complicated than the CIFAR-10/FMINST/MNIST classification task. We added a comment on neuro tasks in the main text and put the result in the appendix.

It is very nice that the authors have thought of the spectral radius as an important parameter regulating the efficacy of their method.

R1-11: We appreciate the positive feedback for our idea.

Ongoing activity. It would seem that the main problem with having error sent at other locations than inference signals is not so much the connectivity itself, as one could indeed think of a relay cell scenario. But the main problem would be if the different cells are engaged in different tasks. It would seem that the authors think that the algorithm would also be robust to that but in my understanding, this has not been shown. The simulations in Figure 5 get close to this, but this would require to explicitly have some of the inference input getting to the error neurons.

R1-12: We think the network would be much more robust and efficient if the error can propagate via different routes, even an indirect route. Moreover, the error can propagate concomitantly with the inference signal on the same RNN, and can share the same path with the error of another tasks, adding to the robustness. Please also see a related reply at R1-17.

As replied in R1-4, we performed additional experiments training two tasks on one network simultaneously (Figure 5b, c). The network can learn both tasks successfully. The error of one task can be seen as perturbation or noise for the other task due to that two dataset are not related, which do not jeopardize learning and even facilitate generalization as found in machine learning (more discussion in Appendix 4).

The article is lacking precision as to how Figure 5a is conducted (how are the input paths chosen? All part of the 'RI' parts?).

R1-13: There are two different groups of RI neurons performing different tasks (RI1 for FMNIST and RI2 for MNIST). We have revised Figure 5 and its caption for clearer descriptions (line 324).

Lack of clarity relating to Figure 4. Line 190, I could not comprehend this section , starting with " We illustrate two concepts of error… (STE)" What is a 'concept of error representation'? They seem to be different type of presentations/dynamics of the error signal, but the terminology 'concept of error representation' seem to indicate something more than that. What are sub-trajectories? Where do these come from? What's being plotted in 3b and 3c? What's X-Y-Z here?

R1-14: We mean two ways of computing the error. One way corresponds to computing the error of neural states between two alternating phases: (1) the network states under input stimulation evolved for certain time steps and (2) the network states under both input stimulation and error perturbation evolved for the same time steps. The starting points for the two phases are the same. This way allows faster training on conventional computer, because the time step can be one. However, it is biologically implausible because the neurons need to go back to the same starting point. The other way corresponds to computing the error of neural states between two stages: (1) the network states under input stimulation evolved for sufficiently many time steps and (2) the network states perturbed by the error for ONE time step. This is much more biologically plausible as explained in R1-8 above.

Sub-trajectory is part of the whole trajectory—the trace of neural states. For instance, OB is the sub-trajectory of OD. We have removed ‘sub-trajectories’ and used clearer wording (line 190 to line 194, line 226 to line 228). Figure 3b visualize the experiment trajectories after principal component analysis, which reduce a 1024-dimension to three dimensions (X, Y, Z). Figure 3c shows the test accuracy with respect to the time steps of neural state evolution, confirming that computing short trajectory error speeds up training. Basically, a minimum of one step for both phases (the 2-2 case) is enough.

Rework Figure 5. The arrangement of Figure 5 is in my opinion unnecessarily difficult to parse. A has one network repeated twice, and b has one network twice as large. These are conceptually very different and the visual symmetry is deceptive. In addition, Figure 5d. It would seem that the curves are superposed but we do not see where.

R1-15: The repeated network in Figure 5a is deleted. And Figure 5d is divided to three sub-figures for clarity.

The discussion was missing a statement about the role of the spectral radius in network. It has been argued as being potentially large, but also as potentially small. So the relationship with performance can imply a trade-off… or not.

R1-16: Yes, there is a trade-off in spectral radius. Too large a spectral radius leads to instability of the network, whereas too small spectral radius renders the network insensitive to the input. We have stressed this point in the main text line 248 to line 250 and line 358 to line 362.

A small note, I thought the paper was not very precise as to the problem of lack of reciprocity in cortical networks. I know the theory of Payeur et al., Naud (2021; NN). is based on mainly the thick tufted, extra telencephalic (ET) L5 pyramidal cells, which are only part of the network L5 network, which makes the question a little more precise, but some precision would be beneficial for the Sacramento et al. Senn NeurIPS Theory as well. These L5-ET cells are thought to be the main part of the L5 network that changes the stimulus representation (Moberg et al. Takahashi 2025, BioRxiv), and these cells are not those receiving reciprocal inputs from higher order inputs (Young, et al. Petreanu 2021, eLife). The other set of cells, the IT cells, seem to be engaged in a reciprocal network between the L5 of the upper layers. The cells also receive feedforward inputs, but they seem less to undergo learning (again the Moberg paper). It would seem to me that this could correspond to an adjoint network where all cells receive the inference signal but only a portion receive the teaching signal.

R1-17: We agree that precisely defining the problem of lack of reciprocity in cortical networks is important. It certainly benefits broader audiences not just in neuroscience.

Our idea is that a cell involved in inference does not have to receive error signal directly from higher hierarchy cells, but can obtain error signal indirectly via other interconnected cells. As long as a cell is interconnected in a local recurrent network, the error perturbation at any point in the local recurrent network can propagate to that specific neuron. This idea is in harmony with others theories based on neural anatomy [Young et al., 2021, *eLife* 10: e59551; Moberg et al., 2025, bioRxiv: 2025.2001.2007.631500], but much more general in its applicability. We have now added a section into the discussion part to make this point clear (lines from 351 to 357).

We appreciate the reviewer’s thoughtful comments.

Reviewer #2 (Recommendations for the authors):The work in this paper should be seen as a stepping stone to something larger. The research space of biologically plausible credit assignment is relatively small (and overall rather under-explored), and this work can be used as a grounding / foundation for future ideas. The authors' demonstration of their methodology's ability to not only change the number of error propagating/receiving neurons (within a multi-region recurrent neural net, i.e., MR-RNN) but also its ability to reuse (neuronal) structures is a nice step forward for computational neuroscience / neuroAI. Beyond the reuse of structures for different tasks, the ability to piece ("compose") together different structures to accomplish the same task shows robustness in the approach's / framework's design.

R2-1: We thank the reviewer for their kind words on our work.

Without detracting from the work accomplished by the authors, there still are a few things that need to be kept in mind when reading/viewing this work. Specifically the base line problems / benchmarks used, i.e., MNIST and FMNIST, are the most basic of baselines / benchmarks. This means that there is no guarantee that this framework/methodology will scale to larger (data)sets such as CIFAR, ImageNet, or other larger natural images (or other high dimensional, more complex collections of data patterns).

R2-2: We appreciate the suggestion. We have performed additional experiments on CIFAR10 dataset as suggested. The test accuracy is 3.5% lower than BP-trained feedforward network, which is reasonably good for a simple two-layer network.

However, as Reviewer #1 pointed out (see R1-10), achieving highest accuracy is not the goal. Our motivation is to demonstrate the capability of a biologically plausible learning mechanism. In this sense, the learning is scalable to deeper structures, as plotted in Figure 4d-e in appendix.

We have now substantially enhanced our discussion on the limitations of AP from line 241 to line 250 and from line 256 to line 258**.**

Please find below a few things that the authors should take care of:1) Small edit on line 362 – there is a typo of "is mall" that I believe should be "is small".

R2-3: Indeed, it is a typo. We have revised it in the revision.

2) All figures could benefit from larger font size/plots; please adjust accordingly to make things more presentable/viewable to readers

R2-4: The font sizes are now increased in all figures.

3) Things to address in the manuscript:a) The authors need to comment on why feedback alignment and adjoint propagation lose accuracy as the model size increase while back propagation gains accuracy (as seen in later results)? What happens if the size of the model increases dramatically, is the accuracy adversely affected? A good of writing should be provided in the main text to address troubling issue and, ideally, a few extra experiments that examine the effect of increasing model size/capacity (this would help to highlight that perhaps, while bio-plausible methods address some limitations of backprop-based DNNs, they might come at the cost of some parameter inefficiency or greater model architecture/parameter complexity).

R2-5: We thank the reviewer for making an important point. The influence of model size on AP is consistent with that on BP and FA.

Indeed, the depth of the model seems to adversely affect the accuracy of AP as shown in Figure 4d-e. However, the accuracies of BP and FA also decrease with increasing number of hidden layers in a fully connected neural network (see Table 1-3 in Reference [Nøkland, 2016, Advances in Neural Information Processing Systems 29]). For deeper networks, BP relies on residual connections and batch normalization to propagate gradient effectively, and FA needs to adapt to DFA to shorten error paths. These measures are in principle also applicable for AP. We have incorporated these comments into the revision from line 246 to line 250 and from line 380 to line 384.

As for the influence of the number of neurons per layer, Figure 4f shows that the network performs better with increasing number of neurons.

b) As of right now this work deals with entirely continuous values, with the RNN being the analog for biological recurrence. Have the authors explored or considered other stateful biological models such as spiking neural networks (SNNs)? If so, how did SNNs preform as compared to the tested model? This is important to address (in writing and, ideally, experimentally) as there is mention of hardware implementations in the Discussion section, since energy-efficiency tends to emerge as a result of these specific styles of networks (in terms of structure), not just from sparsity. At the very least, some consideration of spiking-driven dynamics should be provided given that continuous RNN models typically look like rate-coded approximations of SNNs (and some SNNs, such as those based on leaky integrate-and-fire cells, follow similar neuronal dynamics/flow).

R2-6: We fully agree that extending the Adjoint Propagation (AP) framework to SNNs is a critical and exciting future direction. We believe the AP framework works for SNN as well, since the dynamics of the RNNs are derived from the rate-coded formulation. We now added a paragraph to the Discussion section (lines from 371 to 375) to address this question. We are grateful for this helpful suggestion.

Reviewer #3 (Recommendations for the authors):This study presented a novel neural network architecture for solving supervised classification tasks, aiming at achieving high biological plausibility. Traditionally, such tasks were solved by multi-layered feed-forward network trained according to the backpropagation (BP) algorithm, i.e., trained by error feedback with the weights that are transposed copies of the (temporally evolving) forward weights (called the symmetric feedback); this was argued to be biologically implausible. More recently, it was shown that comparable performance could be obtained even if the symmetric feedback is replaced with fixed random feedback, which is biologically more plausible, because the forward weights become aligned to the feedback weights (called the feedback alignment (FA)). This study examined an architecture (named the Adjoint Propagation (AP)) deviated from the feed-forward network, while using FA. Specifically, whereas the feed-forward network consists of sequentially connected layers, each of which consists of neurons without mutual connections, AP consists of sequentially connected modules, each of which consists of neurons with mutual connections that are learnable (i.e., recurrent neural network (RNN)). Each module consists of four functional blocks: receiving inference signal (RI), sending inference signal (SI), receiving error (RE), and sending error (SE), and interaction across modules (and also between the input and output) are selectively done according to these prespecified blocks. The authors derived update rules for across-module and within-module weights according to a gradient descent, with approximations. Different from BP and in the same manner as FA, the error feedback weights were fixed, and forward weights aligned to them. So in this regard, AP is biologically more plausible than BP and has the same level of plausibility as FA. However, the module of AP has recurrent connections whereas the layer of the feedforward network (used in FA) does not, and in this regard AP can be biologically more plausible than FA, as the authors argue (but see my comment on this below). As for performance in the examined two tasks (using the public data set), AP is largely comparable to (but can be slightly inferior to) FA and BP.

R3-1: We thank the reviewer for an insightful summary of our work. Feedback alignment uses random weights to feedback error, which has made a leap in biological plausibility compared to BP. AP has used the idea of FA, but is centered on modular recurrent neural networks. It allows concomitant propagation of error and inference on the same RNNs and flexible signal routing for credit assignment, which, in our mind, represents another leap in biological plausibility. Please also see R1-1, R1-17 and R2-1.

Strengths:1) The proposed architecture, sequentially connected modules, each of which has functional blocks and learnable recurrent connections, looks indeed closer to a network of multiple cortical areas than the conventional feed-forward network. Although the assumed structure is somewhat artificial (four prespecified functional blocks) and not exactly the same as the actual structure of cortical area, the proposed model can be a good model of cortical information processing.2) Error representation by a difference between two trajectories of neural states in the recurrent module is an interesting idea, and because it can be calculated far before the iteration of weight update becomes converged, it can in principle speed up learning.3) Reuse of the same network for multiple different tasks is an important thing to examine, and the result looks potentially interesting.

R3-2: We thank the reviewer for appreciating the biological plausibility of our framework, its potential in speeding up learning, and its ability to support multi-tasking.

Weakness:1) Whether the representation of error by a difference between two trajectories of neural states can be implemented in a biologically plausible manner seems unclear, as the authors mentioned in Line 198-200. Because the primary focus of this study is biological plausibility, this point is a severe limitation.

R3-3: We fully understand the reviewer’s concerns. We explained the error representation in detail in preceding reply (see R1-8 and R1-14). Basically, the AP framework allows two ways of error computing. One way is to compute error in two alternating phases, which is algorithmically fast but lacks biologically underpinning (i.e. need to store initial states). The other way is to compute the error in two stages: first the sensory input is presented to the network, and then the prediction error propagates through the recurrent network. The input is always online, and the learning can be triggered by the error. In this case, the learning is local in time, which is biologically plausible.

We have revised the text to clarify this point (see line 90 to line 92 and line 204 to line 212).

2) In terms of performance (Table 1), although AP can be said to be largely comparable to FA and BP, precisely speaking, AP is slightly inferior to FA, which is slightly inferior to BP. It is unclear this slight difference in the performance could be more prominent in other tasks (e.g., those for which FA was shown to perform worse than BP).

R3-4: Table 1 provides further experiment on CIFAR10. For more complex task, the performance gaps appear to be larger. The possible reason is that the error is propagating backwards through the Jacobians of randomly connected RNNs. Such random feedback of error is essentially the same as FA and likely inherits FA’s inferiority to BP, and this longer feedback path than FA leads to more inaccurate error propagation in shallow structures (see also R1-3 and R2-5 above).

Our goal is to introduce a novel mechanism of credit assignment (R1-10 and R2-2). As long as the network can learn to perform the tasks, the goal is achieved. However, how to improve performance is an intriguing question for future study. We have added more discussion on this point (see line 142 to line 144 and line 380 to line 384).

3) As for the reuse of the same network for multiple different tasks, comparison with other models is lacking, and it is difficult to judge how good the shown reusability of AP is.

R3-5: There are indeed other works that use the same network to perform multiple tasks [Ruder, 2017; Thung and Wee, 2018, Multimedia Tools and Applications 77: 29705-29725]. Those multi-tasking methods have to do with sharing feature representations (Fang et al., 2017). In contrast, AP framework goes beyond to allow concomitant propagation of inference and error signals of different categories of tasks through the shared network.

We have now incorporated this point in the Discussion section (lines from 365 to 370)

4) A prominent advantage of recurrent neural network, in general, is that it can represent and learn a temporal sequence of information such as language. But this study examined static classification tasks (MNIST dataset: handwritten digits, and FMNIST dataset: grayscale images of 10 clothing categories), and whether the proposed AP can also learn the temporal sequence appears to be unclear.

R3-6: We thank the reviewer for raising this important point. We have now added a paragraph in the Discussion section (lines from 375 to 378) that discuss the potential of AP in learning temporal sequences. In light of the RFLO algorithm, we believe AP can also learning temporal sequence and would like to pursue this line of research in our future work.

5) In the Abstract and Introduction, the authors pointed out the biological implausibility of BP, and introduced AP as a more plausible architecture, while FA was not cited until the middle of the Results. But because FA is an established model that is more plausible than BP and moreover the authors' AP actually also uses FA as a learning principle, it should be fairer to introduce FA in the Abstract/Introduction and framed AP in comparison with FA rather than (or in addition to) BP.

R3-7: We appreciate the advice and have introduced FA in the Abstract/Introduction (see line 41 in the revision). We have also accentuated the comparison with FA.

6) In the originally introduced setting of AP, separate phases for inference and learning are required, and its biological plausibility is unclear. The same problem also exists for BP and FA, but AP requires learning of recurrent network, which usually requires a certain number of iterations, and so the plausibility problem can be said to be severer for AP with this setting than BP or FA. The error representation by a difference between trajectories can beautifully resolve this issue, but it has plausibility issue on its own as I commented in weakness 4).

R3-8: The two phases only distinguish whether the labels are provided to the network, as clarified in R1-8.

Indeed, recurrent dynamics requires certain time to converge. However, we believe this does not necessarily imply a plausibility issue. After all, biological neural networks feature multi-scale recurrent connections (see also R1-6 above), and physical systems converge naturally and fast.

To make the point clear, we have added this comment to the revision at lines from lines from 48 to 54 and line 116.

7) In Figure 3c, cases with more iterations appear to have lower accuracy. Why is it?

R3-9: Yes, the lowering is more prominent with larger spectral radius. For short-trajectory error (STE), the error is represented by the distance between states with only input stimulation and states with input stimulation and error perturbation. When the spectral radius is around and above 1, the neural states become more unstable, and small perturbations gets amplified in long runs.

We now discuss this point explicitly at lines from 232 to 234 in the revision.

8) In Figure 4d, even if the number of RE neurons was set to fairly small (and the network became closer to reservoir computing as the authors mentioned), the performance remained largely comparable. The authors' description sounds it can be a good feature, but doesn't it indicate that the merit of learnability of recurrent connections (as opposed to reservoir computing) is rather limited (even though the performance was better than EP and so there was at least some merit)?

R3-10: We have performed a study on how the number of feedback neurons affect learning performance, which is now included in the main text. When the number of feedback neurons drops below 10, the performance begins to decreases notably. We attribute this transition point to the nature of the classification task: if the prediction/target is encoded in a 10-dimension vector space, the error needs to be of comparable dimension to steer the network efficiently. In short, the number of RE neuron cannot be too small. In our opinion, this number is more correlated to the dimension of prediction error rather than the learnability of RNNs.

And if so, is the reason for it the approximation (Line 369-371) made in the derivation of update rule for recurrent connections, in that this approximation is similar to the omission of non-local terms done in RFLO (Murray, 2019, eLife) or e-prop (Bellec et al., 2020 Nat Commun), which was shown (in the case of RFLO) to degrade the performance even more than random feedback (compared with symmetric feedback) did (Figure 2d of Murray, 2019)?

R3-11: For static image classification task, the approximation made in the derivation of update rule have little influence on performance as shown in Appendix 1-figure 1 in the revision. This approximation has been adopted in previous work [Melchior and Wiskott, 2019; Ororbia and Mali, 2019, Proceedings of the AAAI Conference on Artificial Intelligence 33: 4651-4658]. The case for temporal sequence task can be different, which we attempt to address in our future work.

9) The authors used the term "mixed selectivity" to refer to selectivity across different tasks. Does this match its original terminology (doesn't it refer to selectivity for multiple features within a task)?

R3-12: We thank the reviewer for pointing out the ambiguity. In ref. [Rigotti et al., 2013, Nature 497: 585-590], mixed selectivity is defined as “single-neuron… tuned to mixtures of multiple task-related aspects”. Mixed-selectivity is considered to be essential for multi-tasking, which is an important aspect of cognitive flexibility [Holk and Mejias, 2024, Current Opinion in Behavioral Sciences 56: 101351].

Our usage was related, but not entirely the same. We used the term to refer to the idea the very same neurons can participate in both inference and error propagation. Moreover, they can be involved in multiple tasks simultaneously. We now deleted this term for clarity (see lines from 294 to 296).